# Translation initiation by the hepatitis C virus IRES requires eIF1A and ribosomal complex remodeling

Zane A Jaafar[1], Akihiro Oguro[2†], Yoshikazu Nakamura[2], Jeffrey S Kieft[1,3*]

[1]Department of Biochemistry and Molecular Genetics, University of Colorado Denver School of Medicine, Aurora, United States; [2]Institute of Medical Science, The University of Tokyo, Tokyo, Japan; [3]RNA BioScience Initiative, University of Colorado Denver School of Medicine, Aurora, United States

**Abstract** Internal ribosome entry sites (IRESs) are important RNA-based translation initiation signals, critical for infection by many pathogenic viruses. The hepatitis C virus (HCV) IRES is the prototype for the type 3 IRESs and is also invaluable for exploring principles of eukaryotic translation initiation, in general. Current mechanistic models for the type 3 IRESs are useful but they also present paradoxes, including how they can function both with and without eukaryotic initiation factor (eIF) 2. We discovered that eIF1A is necessary for efficient activity where it stabilizes tRNA binding and inspects the codon-anticodon interaction, especially important in the IRES' eIF2-independent mode. These data support a model in which the IRES binds preassembled translation preinitiation complexes and remodels them to generate eukaryotic initiation complexes with bacterial-like features. This model explains previous data, reconciles eIF2-dependent and -independent pathways, and illustrates how RNA structure-based control can respond to changing cellular conditions.

*For correspondence: jeffrey.kieft@ucdenver.edu

Present address: †Department of Molecular Biology, The Jikei University School of Medicine, Tokyo, Japan

Competing interests: The authors declare that no competing interests exist.

## Introduction

Translation initiation requires a messenger RNA (mRNA) placed in the small ribosomal subunit's decoding groove with the correct start codon paired to the anticodon of a charged initiator methionine tRNA (Met-tRNA$_i^{Met}$). In eukaryotes, the canonical process to accomplish this requires >12 eukaryotic translation initiation factor (eIF) proteins (*Jackson et al., 2010*). Briefly, after recognition of the N7-methylguanosine cap, a set of eIFs recruits the 43S complex that contains the 40S ribosomal subunit, the ternary complex (TC; this contains eIF2, GTP and Met-tRNA$_i^{Met}$) and several other eIFs. This is followed by eIF1- and eIF1A-dependent scanning of the mRNA, start codon identification, hydrolysis of GTP, release of many eIFs, and eIF5B-mediated 60S ribosomal subunit joining to form an 80S ribosome (*Hinnebusch, 2011*; *Martin-Marcos et al., 2011*; *Pestova et al., 2000*; *Unbehaun et al., 2004*). Translation initiation in eukaryotes can also occur by internal ribosome entry site (IRES) RNAs (*Jackson, 2005*). IRESs of viral origin use a subset or none of the eIFs and some bind directly to the ribosome (*Filbin and Kieft, 2009*; *Thompson, 2012*), thus positioning the start codon without a 5' cap or scanning. IRESs are important in viral infection and are also key players in regulation of gene expression, but the full repertoire of strategies they use is not completely understood.

Hepatitis C virus (HCV) contains one of the first discovered and characterized IRESs (*Tsukiyama-Kohara et al., 1992*) and is the prototype of the structurally similar type 3 viral IRESs, found in some *Flaviviridae* and *Picornaviridae* (*Hellen and de Breyne, 2007*). The HCV IRES forms an extended structure, containing two major structural domains (*Figure 1—figure supplement 1A*) (for review:

*Lukavsky, 2009*; *Fraser and Doudna, 2007*). Domain III (dIII) is the largest and provides the affinity for direct binding to the 40S subunit and interactions with eIF3; it contains multiple stem-loop elements emerging from several junctions. Domain II (dII) is an extended stem-loop that docks on the 40S subunit in the vicinity of the E site. The various domains of the IRES work together to recruit the translation machinery and manipulate it to begin protein synthesis.

Mechanistic models for HCV IRES-driven translation have been mostly developed using biochemical approaches with reconstituted systems (*Ji et al., 2004*; *Nomoto et al., 1995*; *Otto et al., 2002*; *Pestova et al., 1998b*; *Sizova et al., 1998*). These studies show that the HCV IRES RNA binds directly to the 40S subunit using several IRES structural domains (*Kieft et al., 2001*; *Lytle et al., 2002*, *2001*), changing the subunit's conformation (*Spahn et al., 2001*). The IRES also binds directly to eIF3 (*Sizova et al., 1998*), which along with the eIF2-containing TC have been described as capable of progressing the HCV IRES preinitiation complex (PIC) to an elongation-competent 80S ribosome (*Pestova et al., 1998b*). These and other studies have pointed to a mechanism in which a naked (unbound by factors) 40S subunit first binds directly to the IRES RNA through interactions with dIII (*Ji et al., 2004*; *Otto and Puglisi, 2004*), placing the start codon into the P site of the decoding groove, followed by recruitment of eIF3 by IRES subdomain IIIb (dIIIb) and association of the TC to position Met-tRNA$_i^{Met}$ to form a 48S* complex (asterisk denotes noncanonical assembly and composition) (reviewed in: [*Fraser and Doudna, 2007*; *Khawaja et al., 2015*]). Although biochemical data suggest a step-wise recruitment of essential translation components, it has also been suggested that the first step in HCV IRES-driven translation could be binding to an assembled 43S PIC (40S pre-bound by other factors) (*Berry et al., 2010*; *Hellen, 2009*; *Jackson et al., 2010*). In both models, it is proposed that subsequent GTP hydrolysis by eIF2, directed by eIF5 and enhanced by IRES dII, induces factor release and subunit joining (*Locker et al., 2007*). Mutations to different parts of the HCV IRES inhibit specific steps in the pathway (*Berry et al., 2010*; *Filbin and Kieft, 2011*; *Kieft et al., 2001*; *Sizova et al., 1998*; *Spahn et al., 2001*) (reviewed in: [*Khawaja et al., 2015*; *Lukavsky, 2009*]), which is proposed to be similar in other type 3 IRESs (*Kolupaeva et al., 2000*; *Pestova et al., 1998b*; *de Breyne et al., 2008*). In addition, the HCV IRES can also initiate translation when eIF2 is inhibited by phosphorylation of its alpha subunit (*Koev et al., 2002*; *Robert et al., 2006*). Under these conditions, it is proposed that 40S subunit and eIF3 binding are followed by eIF2-independent delivery of Met-tRNA$_i^{Met}$ facilitated by eIF5B (*Terenin et al., 2008*), or perhaps by less well-understood eIFs 2A or 2D (*Dmitriev et al., 2010*; *Kim et al., 2011*).

These models for the HCV IRES mechanism provide important insight, but it is still not certain which of the two pathways outlined above is most valid, and several aspects of the mechanism remain unclear. First, although the 40S subunit, the eIF2-containing TC, and eIF3 have been deemed necessary and sufficient for IRES 48S* formation for some type 3 IRESs (under unstressed conditions), the inclusion of other eIFs in toeprinting experiments appears to affect the PIC formed on certain IRESs (*Kolupaeva et al., 2000*; *Pestova et al., 1998b*; *de Breyne et al., 2008*). Specifically, added eIF1 appeared to destabilize 48S PICs formed on the classical swine fever virus (CSFV) IRES (*Pestova et al., 2008*), and on the Simian picornavirus type 9 (SPV9) IRES this factor appears to alter the conformation of the PIC and destabilize tRNA binding (*de Breyne et al., 2008*). When eIF1A was added to reconstituted assembly reactions with the SPV9 IRES, it resulted in a toeprint consistent with 48S stabilization and stabilized binding of tRNA in the absence of eIF2 (*de Breyne et al., 2008*). However, the full mechanistic implications of these observations has not been fully explored. Next, recent cryo-EM structures of eukaryotic PICs (*Llácer et al., 2015*; *Quade et al., 2015*) show overlap between the binding site of eIF2 (within the TC) and HCV IRES dII on the 40S subunit (*Figure 1—figure supplement 1B and C*). This clash could be resolved by the fact that IRES dII can adopt other positions (*Yamamoto et al., 2014*), but still the position of the TC on the 40S subunit blocks access to the decoding groove. Thus, if the IRES bound a preassembled 43S complex it is not clear how the coding RNA would dock into the P site. There is also evidence that dII must be in position to help dock the coding RNA in position (*Filbin and Kieft, 2011*), but if the TC was bound, dII would be excluded from performing this role. These observations argue against a model in which the IRES recruits a 43S complex with the TC already bound. However, the alternate mechanism requires free 40S subunit and unbound eIF3, which is problematic because terminating ribosomes are quickly recycled to 43S complexes; 40S subunits and eIF3 may not exist in sufficient quantities to support this pathway (*Asano et al., 2001*). Finally, a recent cryo-EM structure of 40S subunit bound to eIF3 and part of a type 3 IRES (Classical Swine Fever Virus, CSFV) show eIF3 positioned in a

different location compared to where it binds to the 40S subunit without this IRES (*Hashem et al., 2013*; *des Georges et al., 2015*; *Simonetti et al., 2016*). The interpretation of this observation depends in part on which of the two aforementioned models is correct. Does the IRES bind preassembled 43S complex and displace eIF3, or does it recruit free eIF3 in a noncanonical way? Finally, as mentioned above, the HCV IRES has the ability to operate when eIF2 is inactivated by phosphorylation of its alpha subunit, yet during infection HCV actively suppresses eIF2α phosphorylation (*Garaigorta and Chisari, 2009*; *Vyas et al., 2003*); why has the virus evolved an IRES capable of eIF2-independent initiation if that condition is suppressed during infection? And, which of the proposed eIF2-independent mechanisms is correct (eIF 5B-, 2A-, or 2D-dependent)?

None of these questions invalidate previous studies, but they illustrate that our understanding of HCV IRES (and other type 3 IRES) function is incomplete. As these IRESs are found in medically and economically important pathogens, they raise possibilities as drug targets (*Davis and Seth, 2010*; *Dibrov et al., 2012*; *Hermann, 2016*) and are useful tools for exploring both IRES function and fundamental principles of translation initiation, so it is important that we fully understand how they work. Therefore, we re-examined the mechanism used by the HCV IRES with approaches designed to complement previous biochemical and structural studies. We discovered that in addition to previously reported factors, eIF1A is also needed for full HCV IRES activity. Our results suggest that eIF1A is in part responsible for the stability of Met-tRNA$_i^{Met}$ binding to IRES-bound ribosomes and acts to recognize the docked HCV IRES AUG start codon. Integrating our results with published data, we present a revised and updated mechanistic model for HCV IRES-driven translation in which the IRES exploits and remodels a naturally occurring 'pre-43S' complex created during ribosome recycling, generating a PIC whose composition and function shares features with the bacterial mode of initiation. Our proposed model is consistent with previous studies, helps to answer some of the aforementioned questions, and reconciles competing mechanisms for HCV IRES function in both eIF2-dependent and -independent modes.

## Results

### Translation preinitiation complexes assembled on the HCV IRES contain eIF1A

As a first step, we re-examined the factors present in HCV IRES-assembled PICs using a pull-down approach with biotinylated and immobilized IRES RNA. Rather than reconstituting PIC assembly with purified components, we incubated the RNA in lysate from Huh 7.5 (human hepatoma) cells to replicate cellular conditions and then used antibodies to detect bound 43S complex components eIFs 3, 2, 1A, 1 and 40S subunit (protein rpS6) (*Figure 1A*). The factor-independent Cricket Paralysis Virus (CrPV) intergenic region (IGR) IRES served as a control (*Wilson et al., 2000*). Visual examination of the blot shows that as expected, the CrPV IGR IRES bound 40S subunit but not any initiation factors. The WT HCV IRES pulled down eIF2 and eIF3 as expected, and also eIF1A, but not eIF1. A control HCV IRES mutant lacking affinity for the 40S subunit (*Kieft et al., 2001*), and with severely diminished translation initiation activity (dIIId_GGG-CCC; *Figure 1B and C*), did not bind 40S subunit, eIF1A, or eIF2 and showed decreased binding of eIF3 (*Figure 1A*). These results thus establish the specificity and validity of our pull-down approach. We next examined the effect of mutating important HCV IRES subdomains on factor binding (*Figure 1B–D*). To augment visual examination, we quantitated blots from multiple experiments using bound rpS6 for normalization. Disruption of domain IIIb is known to decrease IRES affinity for eIF3 and translation initiation activity (*Buratti et al., 1998*; *Sizova et al., 1998*); as expected, mutant dIIIb_trunc decreases the amount of eIF3 pulled down. Removing dII (ΔdII) or altering its apical loop (dIIb_ΔGCC) is known to affect PIC formation by changing 40S conformation, inhibiting the proper loading of coding RNA in the decoding groove and decreasing overall translation efficiency (*Filbin and Kieft, 2011*; *Locker et al., 2007*; *Pestova et al., 2008*); as expected they show little or no decrease in bound eIFs 2 and 3. Examining the effect of these mutations on eIF1A binding, both dIIIb_trunc and dIIb_ΔGCC had only a small decease in bound eIF1A, while ΔdII showed a greater decrease. Although both these mutants alter the conformation of the 40S subunit, they do so in different ways and have different functional effects (*Spahn et al., 2001*; *Filbin et al., 2013*); the difference in eIF1A binding to PICs formed on ΔdII compared to dIIb_ΔGCC may reflect this. Overall, these results show that the binding of eIFs

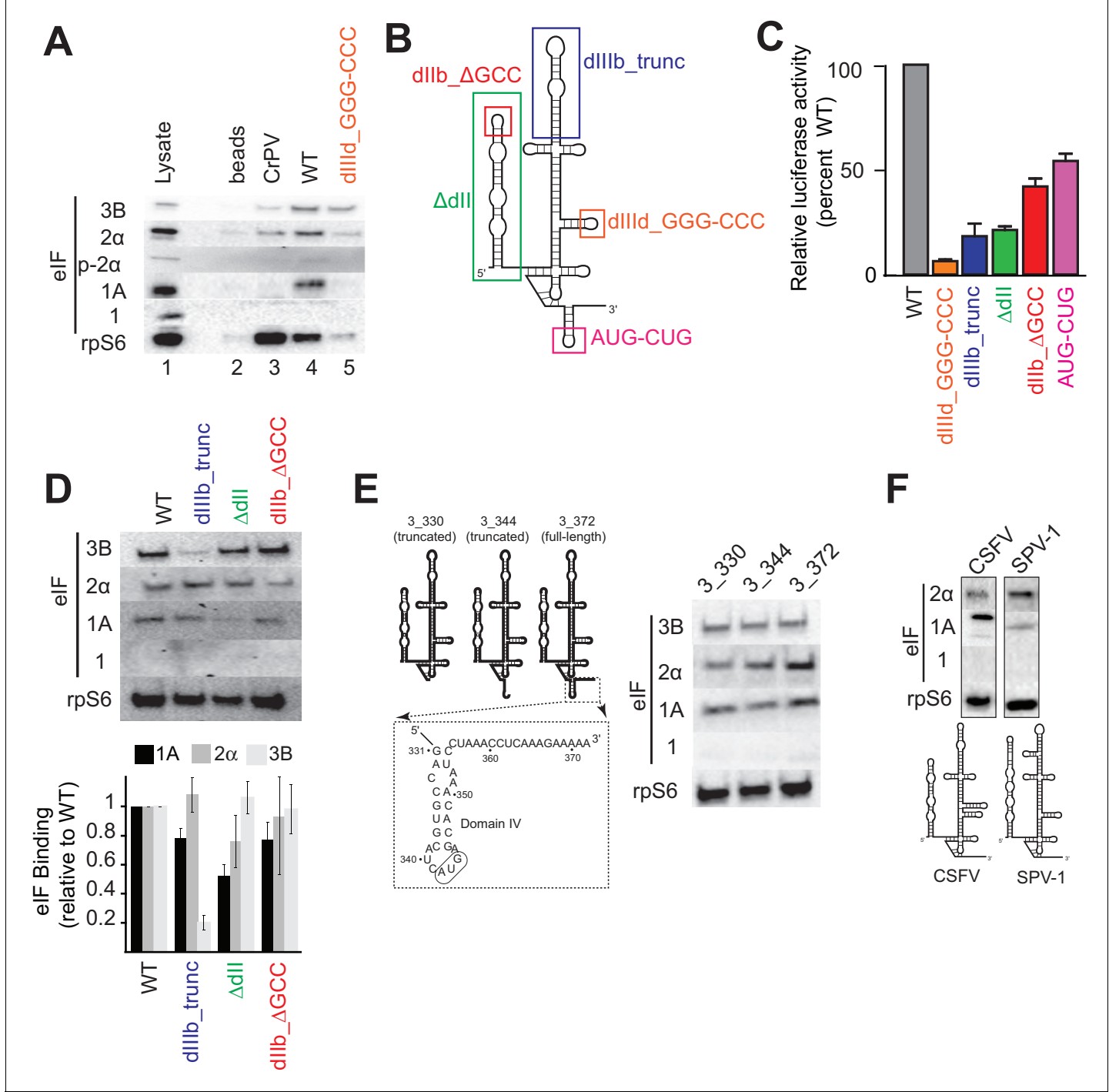

**Figure 1.** Analysis of HCV IRES preinitiation complex composition, formed in lysate. (A) Results of pull-down experiments. HCV IRES was 5' biotinylated and immobilized on ' streptavidin agarose beads. Factors and subunits that bound to the IRES after incubation in lysate were detected by western blot analysis. Lane 1: Input lysate. Lane 2: streptavidin beads only. Lane 3: CrPV IGR IRES RNA control. Lane 4: WT HCV IRES RNA. Lane 5: HCV IRES mutant dIIId_GGG-CCC. (B) Secondary structure cartoon of the HCV IRES RNA, with the location of mutations indicated. (C) Translation initiation activity of the HCV IRES, WT and mutants, with RNA transfections of Huh 7.5 cells. Error bars represent averages ±SEM of ≥3 independent experiments. (D) Pull-downs as in panel (A) with HCV IRES mutants shown in panel (B). Blots were quantitated by densitometry, and the intensity of each eIF band was normalized to the rpS6 band. Graph shows the average of three independent experiments with the amount of each eIF bound to WT set to 1. Error bars represent one standard error from the mean. (E) Pull-downs as in panel (A) with truncated HCV IRESs. Cartoon diagrams of these three truncations are shown, as is the portion of the IRES in which these truncations were made. Visual examination shows a similar qualitative pattern. (F) Pull-downs as in panel (A) with the IRESs from CSFV and SPV-1, along with corresponding secondary structure cartoons.

*Figure 1 continued on next page*

*Figure 1 continued*

The following figure supplement is available for figure 1:

**Figure supplement 1.** HCV IRES structure and clash between HCV IRES domain II and eIF2.

depends on IRES binding to the 40S subunit and the effect of these mutations on the binding of eIFs 2 and 3 is consistent with published biochemical studies. Furthermore, these results also suggest that eIF1A is present on these complexes because it associates with the 40S subunit and that the IRES plays some role in its binding stability, suggesting it may be an important functional part of the HCV IRES-assembled PIC.

During canonical eukaryotic scanning-dependent initiation, eIF1A is important in AUG start codon recognition, interacting with mRNA in the decoding groove. Therefore, we tested whether eIF1A's association with HCV IRES PICs depends on the presence of RNA in the decoding groove (*Figure 1E*). 3′ truncations of the HCV IRES RNA to nucleotide (nt) 344 (3_344; ends after the 'AUG' in the P site), and to nt 330 (3_330; contains no RNA in decoding groove) did not qualitatively change the pattern of factor binding compared to an IRES with 10 codons downstream of the start AUG (3_372) based on a visual examination. This is consistent with eIF1A being recruited through 40S subunit binding to the IRES and not by the protein-coding portion of the IRES RNA.

The presence of eIF1A in HCV IRES PICs led us to test two other type 3 IRESs for eIF1A binding: the aforementioned CSFV IRES and the simian picornavirus type-1 (SPV-1) IRES. eIF1A associates with these IRESs and as with HCV, there is no binding of eIF1 (*Figure 1F*). However, a visual inspection reveals interesting differences in the amount of eIF1A pulled down by these IRESs, hinting at differential use of this factor. These results further support the conclusions that eIF1A is recruited to these IRES PICs through the 40S subunit and that IRES RNAs modulate the stability of eIF1A binding.

## IRES translation depends on the presence of eIF1A

Previous studies using reconstituted PIC assembly assays monitored by toeprinting revealed that eIF1A (in the presence of eIF2, 3, and/or 5B) may only moderately enhance the stability of, and is dispensable for, 48S or 80S complex assembly on type 3 IRESs; but, any stabilizing function of eIF1A is completely negated by concurrent addition of eIF1 (*Fraser et al., 2009*; *de Breyne et al., 2008*; *Pestova et al., 2008*). As a result, eIF1A has not been considered part of the set of factors necessary and sufficient for HCV IRES function (*Fraser and Doudna, 2007*). Indeed, one explanation for the presence of eIF1A on HCV IRES PICs is that it is associated with the 40S subunit as an artifact of ribosome recycling, perhaps associated with eIF3 (*Pisarev et al., 2007*), and thus has no functional role in HCV IRES initiation. We explored eIF1A's functional importance using an RNA aptamer (α-eIF1A) that removes eIF1A from initiation complexes (*Figure 2A*), generated in a manner similar to an eIF4G-binding aptamer (*Miyakawa et al., 2006*). The aptamer was validated using surface plasmon resonance and RNA-protein binding experiments (*Figure 2—figure supplement 1*). To verify that the aptamer could deplete eIF1A from lysate, we performed a pull-down similar to that shown in *Figure 1* using lysate pretreated with either an anti-sense (AS) negative control aptamer or with α-eIF1A. Visual examination of the blot shows that treatment with aptamer did not affect 40S subunit binding to the IRES but significantly reduced the amount of eIF1A bound (*Figure 2B*). This effect was specific, as binding of eIFs 2 and 3 were not affected, and eIF1 remained unbound (*Figure 2B*). Aptamer treatment also reduced eIF1A binding on the CSFV and SPV-1 IRESs, although in the latter case the amount of bound eIF1A is already low, as noted above (*Figure 1F*). There is also visible reduction in protein levels from both cap-driven (Rluc) and HCV IRES-driven (CAT) ORFs of a bicistronic reporter using rabbit reticulocyte lysate (RRL) treated with α-eIF1A compared to lysates treated with control RNAs (*Figure 2C*). This result preliminarily suggested that in this lysate the HCV IRES requires eIF1A for full function. To more fully test the functional importance of eIF1A, we used firefly luciferase (Fluc)-based monocistronic reporters (*Figure 2D*) in quantitative translation assays in both RRL and human cell (HeLa) lysate. Treatment of RRL (*Figure 2E*) or HeLa cell lysate (*Figure 2F*) with α-eIF1A decreased translation of a capped reporter compared to treatment with

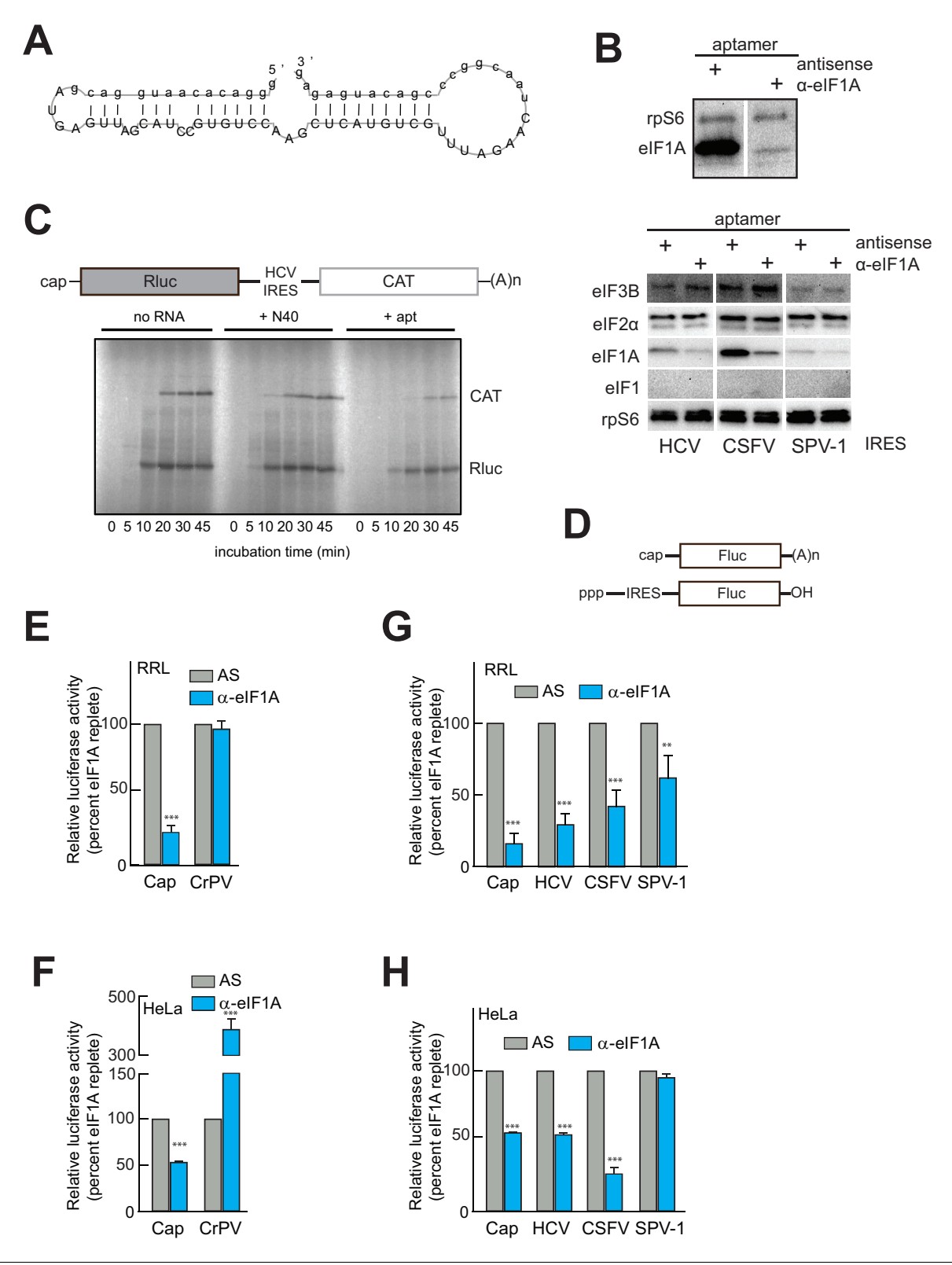

**Figure 2.** An RNA aptamer targeting eIF1A for inactivation negatively affects both cap- and IRES-driven translation in lysate. (A) Sequence and predicted secondary structure of the α-eIF1A aptamer used in this study. (B) Top: Results of pull-down experiment with the HCV IRES (similar to *Figure 1*), showing substantial depletion of eIF1A with aptamer treatment compared to negative control antisense (AS) aptamer RNA. Bottom: Pull-down experiments with several IRES RNAs, with blotting to detect eIFs that should not be removed or depleted by the aptamer. (C) $^{35}$S-Met labeling of

*Figure 2 continued on next page*

*Figure 2 continued*

translation products in RRL using a dual-reporter mRNA template. The aptamer treatment reduces protein levels over time from IRES-driven (CAT) and cap-driven (Rluc) messages. The effect of apatamer (+ apt) addition is shown compared to the addition of no RNA or a randomized 40 nucleotide RNA (+ N40). (D) Diagram of monocistronic reporters used in the experiments of panels (E–H). (E) Results of translation assays in RRL using reporter RNAs. The level of each RNA in untreated lysate is set at 100% and the effect of aptamer treatment is reported as a percentage of that for each RNA. Capped (Cap) RNA is the positive control and the CrPV IRES is the negative control for a requirement for eIF1A. (F) Identical to panel (E) but assayed in HeLa cell extract. (G) Translation assays in RRL using reporter RNAs with the HCV, CSFV, and SPV-1 IRESs. Cap is included as a control. (H) Identical to panel (G) but assayed in HeLa cell extract. In panels (E–H), error bars represent averages ±SEM of ≥3 independent experiments. Statistical significance shown by: *p<0.05, **p<0.01, ***p<0.001.

The following figure supplement is available for figure 2:

**Figure supplement 1.** Validation of α-eIF1A aptamer.

AS, while CrPV IRES-driven translation either did not change or increased with eIF1A depletion. The behavior of these controls is expected, as cap-dependent translation requires eIF1A, while the CrPV IRES is known to be eIF-independent. Examination of the effect of eIF1A depletion on HCV, CSFV, and SPV-1 IRESs revealed variation in the functional requirement for eIF1A in all three. In RRL, all three IRESs showed a decrease in translation with aptamer treatment, with the SPV-1 IRES showing the least and HCV showing the most (*Figure 2G*). In HeLa cell lysate, eIF1A depletion reduced translation initiation by the HCV and CSFV IRESs, but not the SPV-1 IRES (*Figure 2H*). It is noteworthy that in the case of the HeLa lysate, the dependence on eIF1A correlates with the amount of eIF1A pulled down (*Figure 1F*). Specifically, CSFV appears especially sensitive to eIF1A depletion, and it pulled down more eIF1A than did the SPV-1 IRES, which is more refractive to eIF1A depletion. Again, this hints at some mechanistic variability between these related IRESs.

eIF1A has N-terminal and C-terminal tails that perform important roles in start codon selection and TC delivery (*Fekete et al., 2007*; *Saini et al., 2010*; *Acker et al., 2006*; *Maag et al., 2006*). To explore the roles of these tails and to ensure that aptamer treatment was not affecting translation in ways unrelated to eIF1A, we developed a depletion and add-back experimental system using the α-eIF1A aptamer (*Figure 2A*), purified recombinant full-length and truncated eIF1A, and dual luciferase reporters (*Figure 3A* and *Figure 3—figure supplement 1A*). Cap-driven translation and the CrPV IRES controls showed dependence on eIF1A as with monocistronic reporters; adding recombinant full-length eIF1A to the α-eIF1A-treated RRL restored much of the cap-dependent translation activity and returned the CrPV IRES to untreated levels (*Figure 3B*). Addition of eIF1A lacking the unstructured N-terminal tail (ΔNTT) increased cap-dependent translation in the absence of aptamer treatment (*Figure 3—figure supplement 1B*), perhaps because some mutations to the NTT of eIF1A in yeast exhibit a 'hyper-accurate' phenotype that could support increased translation from a reporter mRNA like the one we use in these studies (*Fekete et al., 2007*). Addition of eIF1A with the C-terminal tail deleted (ΔCTT) virtually abolished translation for all initiation mechanisms tested (*Figure 3—figure supplement 1C*). This truncation is expected to enhance the eIF1A-40S subunit interaction (*Maag et al., 2006*; *Fekete et al., 2007*) and may inhibit translation by competing with WT eIF1A and preventing eIF5B-mediated release after subunit joining (*Fringer et al., 2007*; *Acker et al., 2006*). Overall, these results validate this eIF1A depletion and add-back system. Applying this approach to translation assays with different type 3 IRESs (*Figure 3C*), we observed that the diminished HCV IRES activity in eIF1A-depleted lysate was partially restored by add-back of full-length or ΔNTT eIF1A, similar to cap-driven translation. The CSFV IRES did not respond as well to the add-back of full-length eIF1A, but showed more recovery with added ΔNTT eIF1A. As noted above, the SPV-1 IRES was generally less sensitive to eIF1A depletion and consistent with this, add-back of eIF1A had a minimal effect. Interestingly, while addition of ΔNTT eIF1A fully restored translation on capped messages, it only partially restored activity with all the IRESs tested; the significance of this is not clear. The degree to which addition of eIF1A to depleted lysates restores translation roughly tracked with dependence on eIF1A.

We tested two HCV IRES mutants previously shown to negatively affect translation to determine their sensitivity to eIF1A depletion and addition. The aforementioned dIIb_ ΔGCC mutant (activity is ~40% of WT, *Figure 1C*) is defective in docking coding RNA into the decoding groove (*Filbin and*

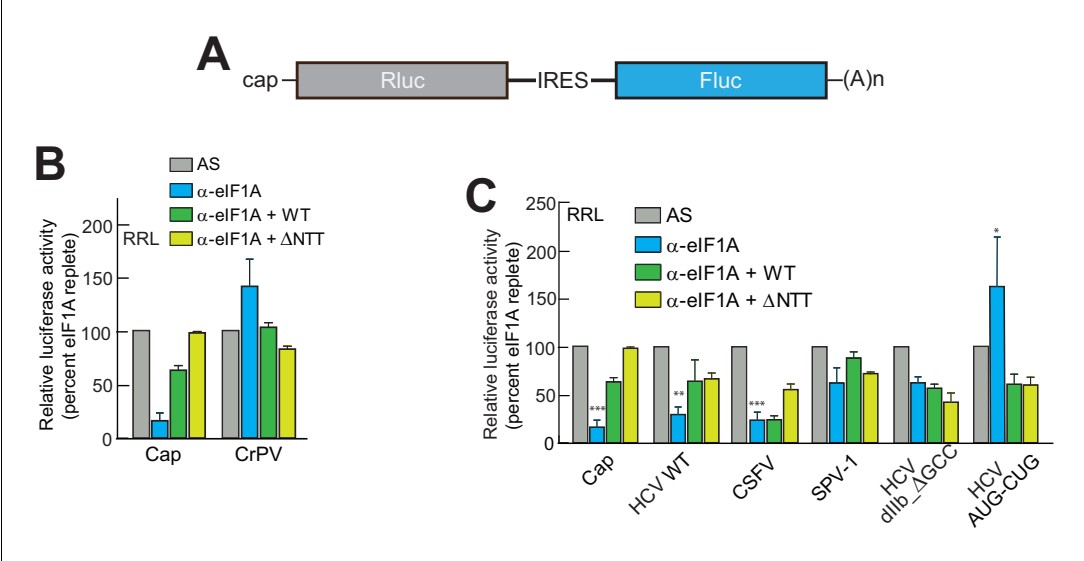

**Figure 3.** eIF1A depletion inhibits translation from the HCV IRES. (**A**) Cartoon showing design of bicistronic dual-luciferase reporter constructs used in the experiments of panels (**B–C**). (**B**) Effect of depletion of eIF1A using the aptamer followed by add-back of WT full-length eIF1A (WT) or an N-terminally truncated mutant (ΔNTT) on translation from the reporter shown in panel (**A**). Experiments were done in RRL. This panel shows the effect on the Rluc (Cap) and a control CrPV IRES. (**C**) Same as panel (**B**) but with WT HCV, CSFV, and SPV-1 IRESs, plus two HCV IRES mutants shown in *Figure 1B*. For panels (B and C), activity in lysate treated with AS control aptamer was set at 100%. Error bars represent averages ±SEM of ≥3 independent experiments. Statistical significance shown by: *p<0.05, **p<0.01, ***p<0.001.

The following figure supplement is available for figure 3:

**Figure supplement 1.** Purity of WT and mutant eIF1A and effects in untreated lysate.

*Kieft, 2011*); it showed some decreased activity with eIF1A depletion but was insensitive to the add-back of eIF1A. This preliminarily suggests that eIF1A's role involves positioning or recognizing RNA in the decoding groove. We therefore tested a mutant with the start AUG mutated to a CUG (AUG-CUG). Start codon mutants are often quite deleterious to translation, but in type 3 IRESs most start codon changes are tolerated to a degree that they allow translation at ~30–50% of wild-type (*Reynolds et al., 1995*). Consistent with this, we observed a decrease to ~50% with mutant AUG-CUG (*Figure 1C*). Interestingly, this mutant's activity was enhanced by the depletion of eIF1A but then decreased with add-back of either full-length eIF1A or the ΔNTT mutant (*Figure 3C*). Overall, these data suggest that eIF1A function on the HCV IRES (and other type 3 IRESs) involves recognition of RNA in the decoding groove, likely inspecting the AUG codon and its pairing with the anticodon of Met-tRNA$_i^{Met}$. These are similar duties to those performed by eIF1A during scanning-dependent start codon selection on canonical mRNAs.

## HCV IRES RNA manipulates preassembled PICs and stabilizes eIF1A binding

The presence and importance of eIF1A on HCV IRES PICs and the fact that eIF1A binds through 40S subunit recruitment supports a mechanistic model in which the IRES binds to preassembled PICs. To explore this further, we developed an approach to recapitulate this type of recruitment event (*Figure 4—figure supplement 1A*). Briefly, we purified 'native' 40S subunit-containing PICs from lysate, incubated them with HCV IRES RNAs, and then these reactions were resolved by sucrose-gradient ultracentrifugation to separate the resultant HCV IRES PICs from factors released upon IRES binding. By examining both the bound and released factor pools, we detected changes in the PICs due to IRES RNA binding. In addition, to explore how stress conditions (and resultant eIF2α phosphorylation) affect these recruitment events, we conducted this experiment both with PICs purified from unstressed cells and from cells treated with DTT, a reductive stressor that leads to robust eIF2α phosphorylation (*Figure 4—figure supplement 1B*). PICs from unstressed 293F cells without added

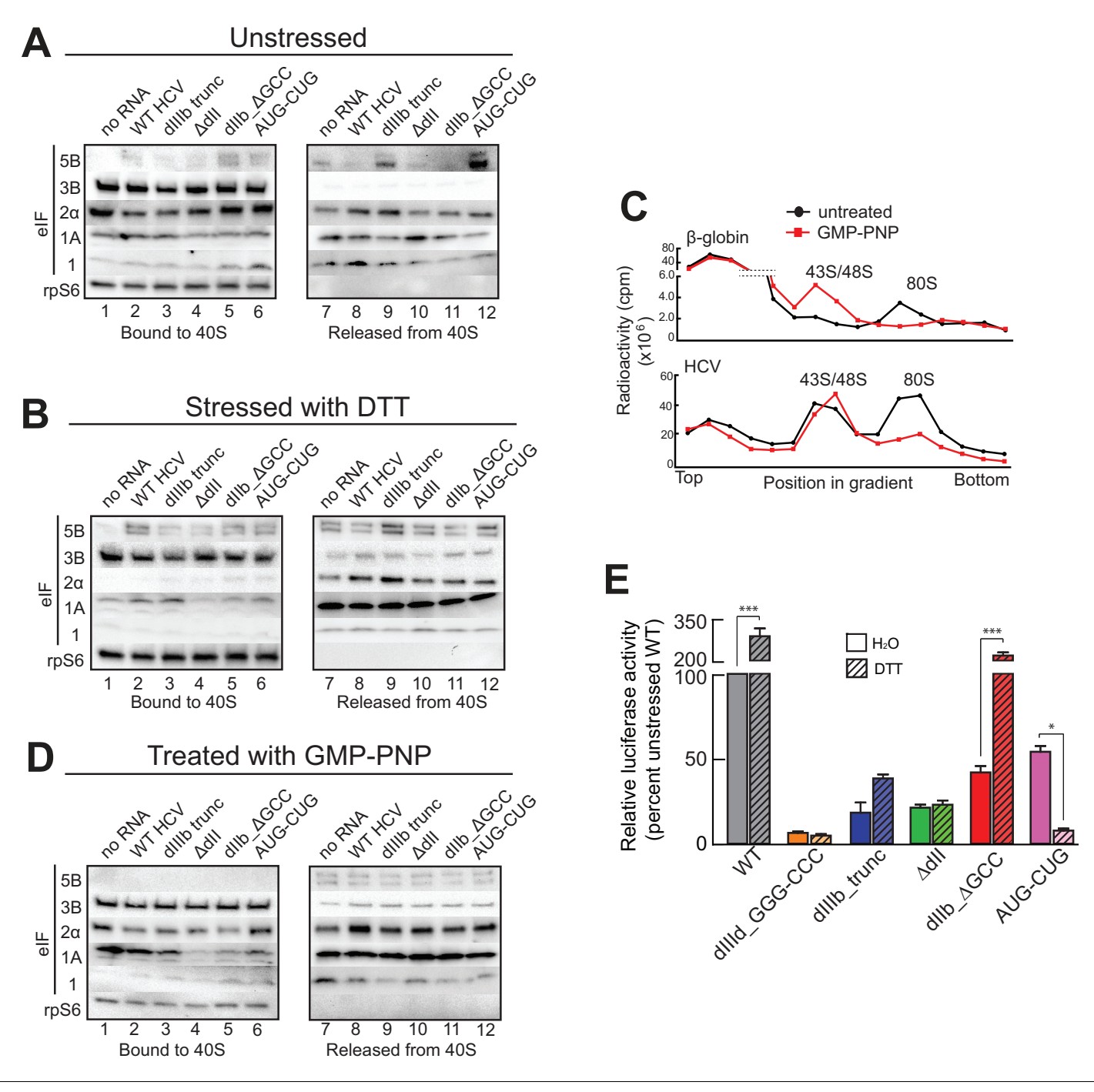

**Figure 4.** HCV IRES binding alters natively purified PICs into complexes suited for IRES-driven translation. (**A**, **B**, and **D**) Results of the experimental protocol shown in *Figure 4—figure supplement 1*. Natively purified 40S subunit-containing complexes were challenged with WT HCV IRES, then the composition of the complex (bound to 40S) and the factors released (released from 40S) were analyzed using western blot analysis. Panel (**A**): PICs from unstressed 293F cells. Panel (**B**): PICs from 293F cells stressed with dithiothreitol (DTT). Panel (**D**): PICs from unstressed cells incubated with non-hydrolyzable GTP analog GMP-PNP. (**C**) 80S ribosome formation on radiolabeled *β*-globin leader or HCV IRES in treated 293F lysate either untreated or treated with GMP-PNP. Reactions were fractionated by ultracentrifugation through a sucrose gradient. Total radioactivity per fraction is plotted. (**E**) Translation from bicistronic reporter mRNA (*Figure 3A*) containing WT or mutant HCV IRES, used to transfect Huh 7.5 cells that were left untreated ($H_2O$ control) or pretreated with DTT. Translation of Fluc is plotted relative to translation of WT HCV IRES reporter in unstressed cells (set to 100%). Error bars represent averages ±SEM of ≥3 independent experiments. Statistical significance shown by: *p<0.05, **p<0.01, ***p<0.001.

*Figure 4 continued on next page*

*Figure 4 continued*

The following source data and figure supplement are available for figure 4:

**Source data 1.** Quantified blot data supporting *Figure 4A–D*.
**Figure supplement 1.** Experimental protocol for analysis of HCV IRES remodeling of natively purified PICs.

IRES contain eIFs 3, 2, and 1A but little or no 5B, as expected (*Figure 4A*, lane 1). There was some loss of all factors from the complex during the centrifugation analysis step (*Figure 4A*, lane 7), with eIF1 the most labile. PICs from stressed cells showed a similar pattern but with dramatic reduction in bound eIF2 (*Figure 4B*, lane 1). There was no phosphorylated eIF2α bound, as expected (*Figure 4— figure supplement 1C*). We verified that the use of these stressors and induced eIF2α had the expected effect on translation using luciferase reporter mRNAs with cap, eIF2-dependent and independent IRESs, observing the expected changes in translation activity (*Figure 4—figure supplement 1D and E*). These results suggest this approach was valid for examining the effect of HCV IRES binding to preassembled PICs. Quantification of western blot data from multiple experiments is contained in *Figure 4—source data 1*.

Adding WT HCV IRES to the purified preassembled PICs had similar effects on PICs from both stressed and unstressed cells. Specifically, eIF2 was displaced, while eIF3 and eIF1A remained bound (*Figure 4A and B*, lanes 2 and 8). Both eIF1A and eIF5B showed increased binding or retention when the HCV IRES was present, more obviously with PICs from stressed cells. Interestingly, there was more eIF5B present overall in complexes from stressed cells. If we use eIF5B's presence as a proxy for a 48S complex prepared to progress to 80S ribosomes, these results suggest: (1) addition of HCV IRES to pre-formed 43S complexes results in PICs capable of progressing to 80S ribosomes, and (2) under stress, the relevant HCV IRES complex can form without eIF2 but contains eIFs 3, 5B, and 1A. Experiments with PICs from unstressed HeLa cells produced similar results (*Figure 4—figure supplement 1F*).

Because GTP hydrolysis by eIF2 is important for changes in canonical 48S PICs, we explored its importance in HCV IRES-driven events by treating the lysate with non-hydrolyzable analog Guanosine 5′-[β,γ-imido]triphosphate (GMP-PNP) before purifying PICs. GMP-PNP exchange was effective enough to prevent 80S ribosome formation on both capped β-globin mRNAs and HCV IRES RNAs (*Figure 4C*). Purified PICs from GMP-PNP-treated lysate contained eIFs 3, 2, and 1A but again no eIF1. Addition of HCV IRES resulted in retention of eIFs 3 and 1A, but there was little or no bound eIF5B on these HCV PICs, as expected (*Figure 4D*, lanes 2 and 8). eIF2 was again displaced from these PICs by the IRES. While the eIF2 release could be a hydrolysis-dependent event due to residual GTP bound to eIF2, the effectiveness of the GMP-PNP exchange suggests otherwise (*Figure 4C*). As outlined in the Discussion section, we do not propose that GTP hydrolysis-independent displacement of eIF2 is a necessary step in the HCV IRES mechanism; however, this result is consistent with the observation that dII and eIF2 occupy the same spot on the 40S subunit, and there may be some competition between the two that must be resolved.

Overall, the results of the experiments presented above suggest the HCV IRES can bind and partially alter pre-formed PICs to achieve a complex capable of 80S ribosome formation using eIF1A, with or without eIF2. The removal of some eIFs could be triggered by IRES binding, or the factors could stochastically dissociate during the incubation and then the IRES could prevent stable re-binding. By either pathway, the end result is a PIC altered by its interaction with the HCV IRES. We used mutants to understand the role of different HCV IRES structural domains in these events and to correlate this with activity in both stressed and unstressed cells. Altering dIIIb (dIIIb_trunc) had a small effect on eIF3 in the complex despite the fact that mutation of this domain dramatically reduces direct eIF3 binding to the HCV and CSFV IRESs (*Figure 4A,B and D*, lanes 3 and 9) (*Kieft et al., 2001*; *Sizova et al., 1998*), again consistent with a mechanism in which eIF3 is recruited by interactions with the 40S subunit. However, a PIC with dIIIb_trunc did not contain eIF5B, which would inhibit progression to 80S ribosomes. This effect is mirrored by the decrease in translation efficiency (*Figure 4E*). Removal of dII (ΔdII) slightly decreased 40S-bound eIF1A but increased retained eIF2 (*Figure 4A*, lanes 4 and 10), again consistent with a potential clash between dII and eIF2 (*Figure 1—*

figure supplement 1B and C). Interestingly, retention of eIF1A on PICs from unstressed HeLa cells was less effected by ΔdII, perhaps reflecting some subtle difference in the makeup or configuration of the PICs from different sources (*Figure 4—figure supplement 1F*). With PICs from stressed cells with this mutant, there is no eIF2 binding (as expected) and eIF1A is less well retained (*Figure 4B*, lanes 4 and 10), while in GMP-PNP-containing complexes eIF1A is not well retained (*Figure 4D*, lanes 4 and 10). Hence, IRES dII may somewhat compete for eIF2 binding (in a position docked in the E site), but it helps keep eIF1A bound. An interesting consequence of dII deletion is that the mutant does not recover activity when cells are stressed (*Figure 4E*), perhaps due to the decreased binding of eIF1A. In summary, both domains II and IIIb contribute to the ability of the HCV IRES to affect the makeup of a recruited preassembled PIC.

We assessed the contribution of HCV IRES elements located in the decoding groove to changing PIC composition by using the dIIb_ΔGCC and AUG-CUG mutants. With PICs from unstressed cells, dIIb_ΔGCC did not cause as much dissociation of eIF2 (*Figure 4A*, lanes 5 and 11), consistent with cryo-EM reconstructions of this mutant which show an altered location of dII (*Filbin et al., 2013*). eIF1A was largely retained. With PICs from stressed cells and dIIb_ΔGCC, there is no eIF1 or eIF2 bound, but eIF1A is present similar to WT (*Figure 4B*, lanes 5 and 11). This correlates with a dramatic increase in activity with stress: dIIb_ΔGCC has inhibited translation initiation in unstressed cells, but more than recovered activity in stressed cells (*Figure 4E*). With AUG-CUG, eIF1A was bound under all conditions, and eIF1 was retained with PICs from unstressed cells and to a certain degree on GMP-PNP-treated PICs (*Figure 4A and D*, lanes 6 and 12). Interestingly, AUG-CUG induces only a 50% decrease in activity in unstressed cells, but strikingly its activity is decreased dramatically when the cells are stressed, dropping to levels similar to the negative control mutant dIIId_GGG-CCC (*Figure 4E*). Thus, the HCV IRES's AUG start codon is much more important for function when eIF2 is depleted. However, the makeup of the complex formed by AUG-CUG with PICs from stressed cells is very similar to that formed with WT (*Figure 4B*, compare lanes 2 and 8 with lanes 6 and 12). The similarities include the presence of eIF5B, suggesting that some unknown step such as tRNA delivery is disrupted by this mutant. Overall, eIF5B binding to PICs from DTT-treated cells is higher when eIFs 2 and 1 are absent and 1A and 3 are present.

## eIF1A affects tRNA$_i$ binding to HCV IRES-bound PICs

Our results thus far show that 40S subunit-bound eIF1A is important for HCV IRES-driven translation. In canonical cap-dependent translation initiation, eIF1A performs roles both during and after scanning. Interestingly, when first discovered, eIF1A was referred to as eIF4C and its first described activity was stimulating tRNA binding to ribosomes (*Schreier et al., 1977*; *Trachsel et al., 1977*). As the HCV IRES does not induce scanning, we hypothesized that eIF1A could be important for 'post-scanning' functions during HCV IRES-driven translation, including stabilizing tRNA$_i$ binding and stimulating subunit joining with eIF5B (*Acker et al., 2006*; *Goumans et al., 1980*). These functions are accomplished cooperatively by the N- and C-terminal tails as mentioned above. We showed that the eIF1A-ΔCTT truncation mutant potently inhibits IRES-dependent translation (*Figure 3—figure supplement 1C*), strongly suggesting that it is important for subunit joining during HCV IRES initiation. To explore if eIF1A is also performing its role of inspecting and stabilizing tRNA$_i$ on these complexes, especially in the absence of eIF2, we combined the approach of purifying and analyzing 'native PICs' from cell lysate with the ability to deplete eIF1A using the α-eIF1A aptamer (*Figure 5—figure supplement 1A*). First, using PICs purified from both unstressed and DTT-stressed cells, we verified that aptamer treatment does not positively or negatively affect binding of eIF2 (*Figure 5A*). The results indicate that any observed effects on tRNA$_i$ levels upon aptamer treatment are independent of eIF2. We then established a baseline for tRNA$_i$ binding to PICs from both stressed and unstressed cells (*Figure 5B*). In the absence of aptamer or IRES, eIF2-bound PICs contain more tRNA$_i$ (assessed relative to 18S rRNA) than do PICs from DTT-treated cells, although the latter retain tRNA$_i$ above background levels, indicating some stably bound tRNA$_i$ to these purified PICs without eIF2.

Using these purified complexes, we examined the effect of depleting eIF1A on tRNA$_i$ association with PICs bound to HCV IRES. Under conditions of eIF2 abundance, treatment with the α-eIF1A aptamer and IRES binding induced no reproducibly statistically significant change in tRNA$_i$ binding (*Figure 5—figure supplement 1B*). However, when we repeated this analysis with PICs from stressed cells, treatment with α-eIF1A aptamer reduced the amount of tRNA$_i$ bound to WT HCV

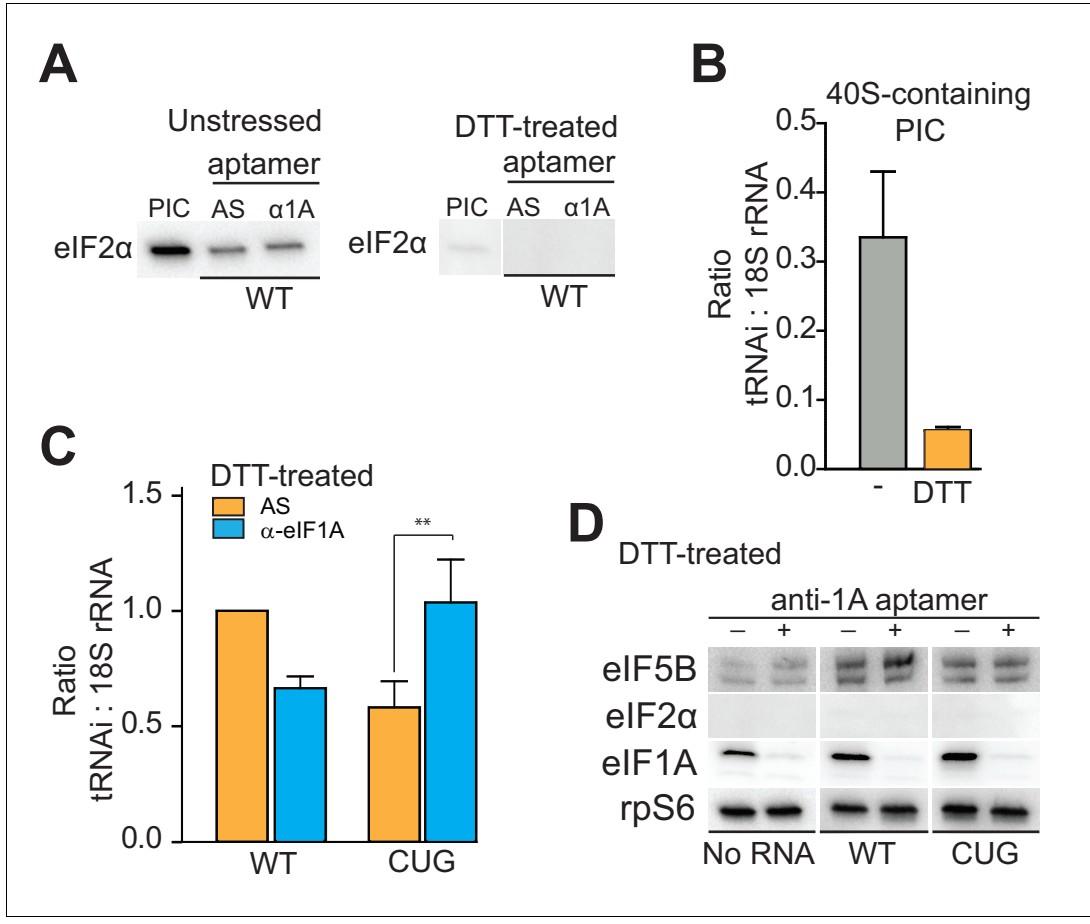

**Figure 5.** eIF1A stabilizes tRNAi^Met binding and inspects the start codon on HCV IRES-bound PICs. (**A**) Western analysis of eIF2 bound to natively purified PICs treated with α-eIF1A aptamer. α-eIF1A aptamer treatment has no effect on eIF2 binding in unstressed or DTT-stressed PICs bound with WT HCV IRES. Input PIC shown for comparison. (**B**) Quantified data from Northern blots measuring the amount of tRNAi bound to PICs from unstressed (grey) or DTT-stressed (orange) cells. PICs were not treated with aptamer or IRES. tRNAi signal presented as a ratio to internal control 18S rRNA signal. Experimental protocol shown in **Figure 5—figure supplement 1**. (**C**) Identical to panel (**B**), but measuring tRNAi binding to PICs treated with AS (orange) or α-eIF1A (cyan) aptamers and bound to WT HCV IRES or the AUG-CUG mutant (CUG). Ratio is expressed as tRNAi compared to 18S rRNA signal, and the signal from untreated, HCV IRES-bound PICs set to 1.0. Error bars represent averages ±SEM of ≥3 independent experiments. Statistical significance shown by: *p<0.05, **p<0.01, ***p<0.001. (**D**) Analysis of factors bound to PICs from DTT-stressed cells, depleted of eIF1A. Western blots to various factors are shown. These are the same samples analyzed in panel (**C**).

The following figure supplements are available for figure 5:

**Figure supplement 1.** PIC preparation strategy for northern blotting quantification.

**Figure supplement 2.** Presence of alternate eIFs 2A and 2D in PICs, and effect of knockdowns.

IRES-containing PICs (**Figure 5C**). Binding of the AUG-CUG mutant to PICs from stressed cells also decreased tRNAi binding, but depletion of eIF1A increased tRNAi binding. Taken together, these data suggest that in the context of HCV IRES translation, eIF1A is performing roles analogous to its canonical roles: it contributes to stable tRNAi binding but is also involved in discriminating against non-AUG start codons, and this role is particularly important during stress conditions. Furthermore, eIF1A depletion and resultant tRNAi binding does not correlate with the presence of eIF5B on complexes (**Figure 5D**, compare to **Figure 4**). Hence, eIF5B and eIF1A are very likely acting as partners

in post-tRNA$_i$ binding subunit association, and while eIF5B is playing an important role in tRNA$_i$ binding and subunit joining, it is not likely acting as a direct tRNA$_i$ delivery agent when eIF2 is depleted.

Finally, as eIF2A and eIF2D have been proposed as possible tRNA delivery agents in the absence of eIF2 (*Dmitriev et al., 2010*; *Kim et al., 2011*; *Skabkin et al., 2010*), we assessed their role. We knocked down either eIF2A and/or eIF2D in Huh 7.5 cells and assessed the effect of HCV IRES translation with and without thapsigargin stress, replicating previously reported conditions (*Kim et al., 2011*). Knockdown of these factors did not affect HCV IRES function under stress (*Figure 5—figure supplement 1A and B*). Based on this result, we tested the effect of eIF1A knockdown in order to verify that a knockdown of a factor important for HCV IRES activity would affect translation by this assay, and also tested the effect of a second stressor, DTT (*Figure 5—figure supplement 2C and D*). Partial knockdown of eIF1A resulted in a modest but reproducible decrease in HCV IRES-mediated translation in unstressed cells and cells stressed with either thapsigargen or DTT, while depletion of eIF2A and eIF2D together had no effect. Note that we achieved complete knockdown of eIF2A and eIF2D during 20 nM siRNA treatment (*Figure 5—figure supplement 2A*) but only achieved partial knockdown of the same factors at 10 nM siRNA concentration (*Figure 5—figure supplement 2C*). We used lower siRNA concentrations when testing eIF1A knockdown because higher concentration siRNA-targeting eIF1A was toxic to cells. Overall, we conclude that eIF2A and eIF2D are not substituting for eIF2, and this experiment also confirmed the importance of eIF1A in a cell-based translation assay.

## Discussion

The type 3 IRESs are found in medically and economically important pathogens and they are important models to explore fundamental tenets of eukaryotic translation initiation. Key questions, paradoxes, and unknowns remain regarding their mechanisms of action, including some debate about how the 40S subunit and initiation factors are recruited to the IRES RNA. This motivated us to explore the mechanism of these IRESs with approaches designed to address key remaining questions.

We have discovered important specific roles for eIF1A in type 3 IRES translation initiation, which mirror its roles in canonical cap- and scanning-dependent initiation. First, it stabilizes tRNA$_i$ binding to HCV IRES PICs. This role makes sense, as eIF1A was initially referred to as eIF4C and its first described activity was stimulating tRNA binding to ribosomes (*Schreier et al., 1977*; *Trachsel et al., 1977*), consistent with a post-scanning role of eIF1A in facilitating tRNA$_i$ stability on proper start codons. A second function mirrors eIF1A's role in discriminating against improper AUG codons and codon-anticodon pairing (*Maag et al., 2006*). As in scanning, this role depends on eIF1A's NTT (*Fekete et al., 2007*), which also promotes scanning along with eIF1 (*Maag and Lorsch, 2003*; *Pestova et al., 1998a*). However, as eIF1 is not present on the HCV IRES-bound PICs, the scanning role of the NTT is likely unnecessary. Third, eIF1A's CTT almost certainly plays a critical role in stimulating eIF5B-bound GTP hydrolysis in subunit joining during HCV IRES-driven initiation (*Acker et al., 2006*).

eIF1A's role in HCV IRES translation initiation is consistent with the interplay of eIF1 and eIF1A with the 40S subunit and other factors during canonical initiation (*Nanda et al., 2013*). Briefly, start codon recognition and GTP hydrolysis on eIF2 result in departure of eIF2 and eIF1 and movement of the 40S subunit into a 'closed' conformation in which eIF1A binding is further stabilized. As previously noted, a 'closed' conformation is reminiscent of that seen when the HCV IRES is bound (*Filbin et al., 2013*; *Fraser et al., 2009*; *Llácer et al., 2015*; *Spahn et al., 2001*), a conformation that depends on IRES domain dII. These links between ribosome conformation, eIF1A stability, and dII suggest an explanation for the reduced occupancy of eIF1A on IRES complexes lacking dII. HCV IRES binding to the PIC may promote a 'closed-like' post-scanning conformation that stabilizes eIF1A binding, promotes tRNA$_i$ binding, and facilitates progression to 80S ribosomes with eIF5B. dII may functionally mimic post-scanning duties of eIF1 and work cooperatively with eIF1A to inspect the AUG. Our observed increase in AUG-CUG mutant translation during eIF1A depletion is consistent with this.

Although our data strongly support a role for eIF1A in type 3 IRES initiation, it is important to consider reports that may initially appear to suggest otherwise. First, it has been reported that

addition of eIF1/1A to HCV and CSFV IRES-bound ribosomes destabilizes the ribosome-IRES complex (*Pestova et al., 2008*). One possible explanation is that PICs generated from purified components and subsequently challenged with eIF1A differ from an IRES recruited to a natively assembled PIC containing eIFs 1, 1A, and 3. Consistent with this, stable eIF1A binding depends on other PIC components (*Maag and Lorsch, 2003*), and a natively preassembled PIC may more readily assume post-scanning conformations that promote stable eIF1A binding. In fact, eIF1A alone does not seem to destabilize 48S* PICs on the CSFV IRES in vitro when added with eIFs 3 and 5B (*Pestova et al., 2008*). Second, it has been reported that eIFs 2 and 3 are deemed necessary and sufficient for SPV9 IRES 48S complex formation (*de Breyne et al., 2008*). There may be differences in the need for eIF1A among different type 3 IRESs, consistent with our observations with the SPV-1 IRES. However, toeprinting studies that monitored reconstituted PIC assembly on this IRES indicated an effect induced by including eIF1A, although this did not lead to a conclusion regarding a functional role. Hence, rather than contradicting these studies, our results build on these findings and expand understanding.

The fact that we find eIF1A bound in a functionally important role requires that we consider how it is recruited to the HCV IRES PIC, in turn helping to discriminate between different HCV IRES mechanistic models. As mentioned above, evidence suggests that ribosome recycling occurs immediately after termination, when eIFs 3, 1, and 1A load on the 40S subunit. In fact, eIF3j (subunit of eIF3) contacts eIFs 1 and 1A along the decoding groove of the ribosome in a coupled binding reaction on yeast ribosomes (*Aylett et al., 2015*), and manipulation of ribosomes to expel eIF3j is a necessary step in IRES binding (*Fraser et al., 2009*), supporting a mechanism of IRES binding to a factor-loaded 43S complex. In our hands, natively purified 40S subunit-containing PICs have a number of eIFs preloaded, including 1A. Thus, we assert that the weight of evidence is consistent with a model in which the HCV IRES binds and manipulates a factor-loaded PIC within a cell, rather than finding and binding free 40S subunit, eIF3, and eIF1A (*Figure 6*).

We favor a model for HCV IRES function that combines aspects of previous models, growing knowledge of eukaryotic translation initiation, and our new data (*Figure 6*). In this model, we propose that the IRES RNA first binds an assembled PIC that is normally generated as a necessary intermediate in the ribosome recycling process (*Pisarev et al., 2007*). This PIC contains eIFs 1, 1A, and 3 but is awaiting binding of the eIF2-containing TC. As binding of the TC is proposed to be a relatively slow step (*Majumdar et al., 2003*), this 'TC-deficient' PIC should exist in sufficient amounts to support HCV IRES translation. Because eIF2 is not present, there is no steric hindrance on the PIC for IRES dII to dock into position and change the 40S subunit conformation and no block for placement of the start codon in the decoding groove. Although purely speculative, it is tempting to wonder if a role of dII is to help the IRES select PICs that do not have the TC bound and hence have an open and available decoding groove. Binding of the IRES dIIIb could then interact with bound eIF3, moving it to a 'displaced position' as seen in recent cryo-EM structures (*Hashem et al., 2013*). We propose that labile eIF1 also departs or is displaced by HCV IRES binding. This altered PIC now has a composition and conformation that facilitates Met-tRNA$_i$^Met binding by one of two modes. If eIF2 is abundant, then the TC can bind, an event that would require IRES dII to move to an alternate position. Although the structure of such a complex has not been solved, this position could be similar to that observed in recent cryo-EM reconstructions of the HCV IRES bound to 80S ribosomes with tRNA (*Yamamoto et al., 2014*). This would be followed by GTP hydrolysis by eIF2 and departure of the factor. Under conditions of eIF2 inhibition, we propose that Met-tRNA$_i$^Met binding occurs directly, without a dedicated delivery factor, and is stabilized by eIF1A and eIF5B. In both modes of tRNAi delivery, eIF1A and eIF5B then work together for the final stages of codon-anticodon verification and subunit joining. We assert that this model is consistent with published results, clarifies discrepancies between existing models, presents new similarities between canonical and IRES-driven initiation, and helps to link eIF2-dependent and -independent modes of IRES initiation; features, implications, and challenges of this mechanism are discussed below.

First, this model provides a purpose for displacing eIF3 from the 40S subunit even though eIF3 is needed for full IRES activity. eIF3 helps recruit eIF1A and 1, and thus may be important for creating the PIC that we propose is the substrate for the HCV IRES. However, partially displacing eIF3 could allow conformational and compositional remodeling of the PIC to include removal of eIF1 but still maintain some key eIF3 functions. Second, the idea that the IRES binds selectively to a PIC that contains eIFs 3, 1, and 1A is appealing because it takes advantage of a known intermediate in the

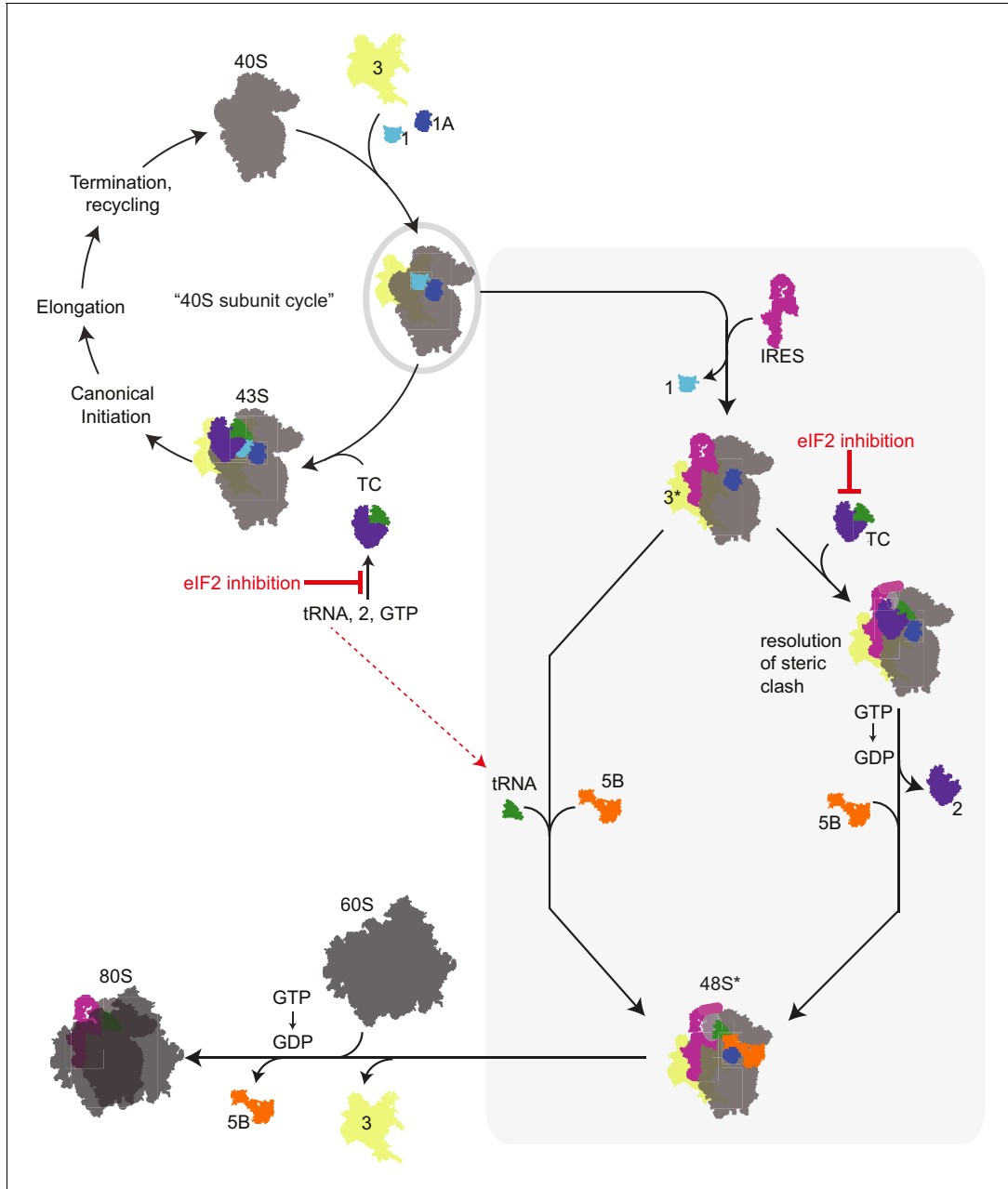

**Figure 6.** Revised mechanism for HCV IRES-driven initiation. This model is described in the main text. Briefly, we propose that the HCV IRES (and probably other type 3 IRESs) recruits a pre-assembled 'TC-deficient' PIC containing eIF1, 1A, and 3 (grey oval), which is an intermediate in the normal 40S recycling process. IRES binding to this PIC partially remodels it to include displacing eIF3 (yellow, '3*') and either passively or actively releasing eIF1. This remodeled PIC is used under both eIF2 active and inactive conditions (grey shaded box). When active eIF2 is abundant, delivery of tRNA through the TC presumably requires movement of IRES domain II to an alternate position (*Yamamoto et al., 2014*) to resolve a steric clash (*Figure 1— figure supplement 1B and C*). If eIF2 is inhibited, this would have the effect of increasing the amount of available 'TC-deficient' PICs and perhaps free tRNA_i. While we cannot prove that an 'alternate' factor is not used to deliver tRNA_i under these conditions, we assert that available evidence supports a mode of tRNA_i recruitment using eIF1A and eIF5B..

normal 40S recycling process that contains factors necessary for IRES activity. By binding this PIC and remodeling it, the IRES creates a complex that can be used in either the eIF2-dependent or -independent modes, and in fact the eIF2-independent pathway could operate in both stressed and unstressed conditions. Although HCV has ways to prevent eIF2α phosphorylation, being able to

operate under a broad set of conditions could confer an advantage to the virus. Third, IRES binding to this preassembled PIC can explain why we observe increases in HCV IRES activity with eIF2α phosphorylation. Specifically, as levels of this TC-deficient PIC increase in response to eIF2 inactivation there would be more available for the IRES to bind. Simultaneously, this would increase the pools of free Met-tRNA$_i^{Met}$ for direct binding to the HCV IRES PIC with assistance from eIFs 1A and 5B. Indeed, mammalian stress granules contain TC-deficient PICs (*Kedersha et al., 2002*), and many viruses (including HCV) modify or co-opt stress granules during infection (*White and Lloyd, 2012*), suggesting that co-opting eIF2-deficient PICs is a mechanism used by HCV during infection.

One question that has been somewhat unresolved is the mechanism by which Met-tRNA$_i^{Met}$ is delivered to the HCV IRES PIC under conditions of eIF2 depletion. One possibility proposed in the literature is that 'alternate' initiation factors eIFs 2A or 2D substitute for eIF2. However, in our hands effective knockdown of these factors did not affect HCV IRES function in stressed cells. Another mechanism proposed that eIF5B is responsible for either delivering Met-tRNA$_i^{Met}$ or stabilizing Met-tRNA$_i^{Met}$ binding to HCV IRES PICs (*Terenin et al., 2008*). We observed Met-tRNA$_i$ binding to HCV IRES-containing PICs that did not correlate closely with bound eIF5B and thus eIF5B may not be the sole responsible factor. Terenin et al. did not identify a role for eIF1A, but it is possible that eIF1A was present in the lysate used for some of their experiments and could be working in conjunction with eIF5B. We assert that a reasonable mechanistic hypothesis for Met-tRNA$_i^{Met}$ recruitment is that eIF1A works with eIF5B to select the Met-tRNA$_i^{Met}$ and stabilize its binding in the absence of a dedicated delivery factor replacement for eIF2. Indeed, while not fully proven, several lines of evidence support the idea of direct Met-tRNA$_i^{Met}$ recruitment to HCV IRES (and by extension, at least some other type 3 IRES) PICs. First, such a mechanism has been observed in vitro with reconstituted studies of the SPV9 IRES, which show that Met-tRNA$_i^{Met}$ can bind to PICs on that IRES independently of eIFs and may be further stabilized by eIF1A (*de Breyne et al., 2008*). Thus, other type 3 IRESs could do the same in cells under conditions of high abundance of free Met-tRNA$_i^{Met}$. As eIF2α phosphorylation results in reduced affinity for Met-tRNA$_i^{Met}$ (*Kapp and Lorsch, 2004*), such conditions could readily exist in stressed cells. Indeed, we observe an increase in IRES translation activity under these conditions. Also, we observe that under conditions of eIF2 inhibition, the HCV IRES is far more sensitive to mutation of the start codon than under conditions of eIF2 abundance. One explanation for this observation is that direct Met-tRNA$_i^{Met}$ binding is more dependent on the codon-anticodon interaction when there is no dedicated delivery factor and eIF1A and 5B alone are inspecting and stabilizing the interaction. This could also help explain the AUG start codon conservation in this group of IRES RNAs despite the fact that they appear to tolerate other codons in conditions of abundant eIF2.

Finally, direct Met-tRNA$_i^{Met}$ binding to ribosomes is accepted in bacterial initiation, thus there is no clear biological reason why it should not occur elsewhere under the right conditions. Indeed, it has already been proposed by Terenin et al. that translation directed by the HCV IRES can operate in a bacterial-like mode using eIF5B to facilitate tRNA binding in a way that mimics its bacterial counterpart, IF2 (*Terenin et al., 2008*). Our data now allow us to extend this analogy by including eIF1A, the functional homolog of bacterial IF1. Although IF1 lacks the CTT and NTT of eIF1A, similar functions have been reported for both factors. Specifically, both occupy similar positions on the ribosome, stabilize tRNA$_i$ binding, and communicate with eIF5B (or IF2) (*Boileau et al., 1983*; *Choi et al., 2000*; *Laursen et al., 2005*). Our model does not include a homolog of IF3. However, one major function attributed to IF3 is prevention of premature subunit association (*Grunberg-Manago et al., 1975*), a function ascribed to eIF3 (*Goumans et al., 1980*; *Siridechadilok et al., 2005*). Hence, the combination of eIF1A, 5B, and 3 with the 40S subunit could comprise a set of factors that parallel IFs 1, 2, and 3 working with the 30S subunit. Combining this observation with the ability of the IRES to bind directly to the ribosome with an AUG placed in the P site, a strategy described as 'prokaryotic-like' (*Berry et al., 2010*; *Pestova et al., 1998b*), the process of translation initiation by the HCV IRES in the absence of eIF2 shares many characteristics with bacterial translation initiation. This has interesting consequences for considering the evolutionary link between bacterial and eukaryotic initiation mechanisms and the role of IRESs in that evolution.

# Materials and methods

## IRES, protein, and plasmid sequences

Hepatitis C virus subtype 1b internal ribosome entry site (HCV IRES) (AB691953) nt 40–372. Classical Swine Fever Virus strain HCLV internal ribosome entry site (CSFV IRES) (AF531433) nt 70–402. Simian sapelovirus 1 strain 2382 internal ribosome entry site (SPV-1 IRES) (AY064708) nt 327–772. Cricket paralysis virus isolate CrPV-3 internal ribosome entry site (CRPV IRES) (KP974707) nt 6025–6246. *Homo sapiens* beta-globin mRNA 5' untranslated region (AF007546) nt 1–79. *Homo sapiens* translation initiation factor 1A X-linked (eIF1A) (AAH00793) coding sequence.

## pDBS

The pDBS vector was assembled using the firefly luciferase gene and backbone from the pGL3-control (Promega, Madison, WI) reporter vector with the upstream *Renilla* luciferase (Rluc) gene cloned from the pRL-CMV (Promega) cloned between the *Hind*III and *Xba*I restriction sites. IRES sequences cloned into the resulting bicistronic reporter vector were cloned between the *Eco*RI and *Nco*I sites and placed in frame with the downstream firefly luciferase (Fluc) coding sequence.

## pUC19

Standard pUC19 (Invitrogen, Carlsbad, CA) was purchased and constructs containing IRES sequence alone or IRES sequences in frame with the Fluc reporter gene were amplified from parent vectors using forward IRES primers with an additional 5' T7 sequence and a reverse primer corresponding to the 3' end of the IRES or the cut site of *Bam*HI in the pDBS vector.

# Primer and oligonucleotide sequences

| Primer name | Sequence 5'−3' |
| --- | --- |
| HCV F | CTCCCCTGTGAGGAACTACTGTCTT |
| CSFV F | CACCCCTCCAGCGACGGCCGAAC |
| SPV-1 F | GTGGTAAGTGATGTTAGTCATTG |
| CrPV F | AGCAAAAATGTGATCTTGCTTGTA |
| CSFV R | GGTATAAAAGTTCAAAGTGATTCAACTC |
| SPV-1 R | CTCCGAGGAGTCATCCTCATAGATTGCCATCTTAGAGAATGTCTT |
| eIF1A_WT F | ATGCCCAAGAATAAAGGTAAAGGAG |
| eIF1A_WT R | TTATCATGCGGCCGCAAGCTTGATGATGTC |
| eIF1A_ΔNTT F | TCTGAAAAAAGAGAACTGGTATTC |
| eIF1A_ΔCTT R | TTATCAAGTTTCATTGATTTTAGCATGC |
| CSFV_ΔdII F | ACTAGCCGTAGTGGCGAG |
| HCV_ ΔdII F | ACCCCCCCTCCCGCCGGGAGAG |
| SPV-1_ ΔdII F | ACACGACCGTACACGAAAG |
| α-eIF1A aptamer | GGGACACAAUGGACGAUGAGUUAGCAUCCGUGUCCAAGCUCAUGUCGUUUAGAACUAACGGCCGACAUGAGAG |
| AS-eIF1A aptamer | CUCUCAUGUCGGCCGUUAGUUCUAAACGACAUGAGCUUGGACACGGAUGCUAACUCAUCGUCCAUUGUGUCCC |
| 18S northern probe | CCATCCAATCGGTAGTAGCG |
| tRNA^Met_i Northern probe | TAGCAGAGGATGGTTTCGATCCATCGACCTCTGGGTTATGGGCC |
| tRNA^Met_e Northern probe | TGCCCCGTGTGAGGCTCGAACTCACGACCTTCAGATTATGAGACTG |
| 5' T7 extension | TAATACGACTCACTATAGGG |
| DBS *Bam*HI R | CCCTAACTGACACACATTCCACAGC |

## Cell lines and cell culture conditions

Huh 7.5 cells were maintained in Dulbecco's modification of Eagle's Medium (DMEM) supplemented with 10% fetal bovine serum (FBS), 10 mM HEPES, and 1X non-essential amino acids (NEAA). Cells were routinely passaged every 48–72 hr and split 1:5. Freestyle 293-F cells were grown in suspension in FreeStyle Expression Medium (Invitrogen). Cells were harvested at $\sim 3.0 \times 10^6$ cells/mL and washed once in PBS before pelleting. Cells were routinely passaged every 48–72 hr and split 1:10. HeLa: Cells were maintained in DMEM supplemented with 10% FBS. Cells were passaged every 48–72 hr and split 1:5. Other cell lines were obtained from other laboratories; the identities of the Huh 7.5, HEK293, and HeLa cells were authenticated by STR profiling. Cells were negative for mycoplasma contamination as tested by PCR.

## Reporter mRNA and IRES RNA transcription

Mono- and bicistronic reporter mRNAs were transcribed using the T7 mMessage mMachine or MegaScript kit (capped vs. uncapped) (Invitrogen) from linearized vector or PCR-generated templates, then treated with the Poly(A)-tailing kit (Invitrogen). IRES RNA constructs were PCR-amplified, then in vitro transcribed using T7 polymerase.

## mRNA transfection and chemical treatment

Huh 7.5 cells were grown in 24-well plates to ~80% confluence, then pretreated with DTT (2 mM) or mock treated for 1 hr. Pretreated cells were transfected with 1 µg bicistronic reporter RNA for 2 hr before being analyzed for luciferase activity. Relative luciferase activity is the percent of WT IRES activity in unstressed cells. Error bars on all translation data represent the average activity ±SEM of at least three independent experiments.

## IRES RNA biotinylation and ribosome pull-down

Wild-type or mutant IRES RNAs were 5' end labeled with biotin using the 5'-EndTag kit (Vector Labs, Burlingame, CA). Huh 7.5 cells were pelleted from 15 cm plates either untreated or pretreated for 3 hr with 2 mM DTT, lysed by syringe and needle, then clarified and stored at 4°C. Ten micrograms of biotinylated IRES was bound to 50 µL streptavidin-agarose bead slurry, washed, then incubated with cell lysate for 30 min at 37°C. After incubation, beads were washed and used for SDS-PAGE and western blotting.

## Antibodies

Antibodies to initiation components were used at the following dilutions according to manufacturer's protocols: eIF1 (1:1000, Cell Signaling Technologies, Danvers, MA), eIF1A (1:500, Santa Cruz Biotechnology, Dallas, TX), eIF2$\alpha$ (1:500, Santa Cruz Biotechnology), p-eIF2$\alpha$ (1:1000, Cell Signaling Technologies), eIF2A (1:1000, ProteinTech, Rosemont, IL), eIF2D (1:1000, ProteinTech), eIF3B (1:500, Santa Cruz Biotechnology), eIF5B (1:5000, Assay Biotech, Sunnyvale, CA), and rpS6 (1:5000, Cell Signaling Technologies).

## Anti-eIF1A aptamer generation and validation by SELEX

We generated RNA aptamers to eIF1A by SELEX (Systematic Evolution of Ligands by EXponential enrichment) (*Ellington and Szostak, 1990*; *Tuerk and Gold, 1990*) from RNA pools randomized over 40 nucleotides (nt). Surface plasmon resonance (SPR) was used to measure binding to eIF1A. The effect of the aptamer in RRL for both cap-dependent and HCV IRES-dependent translation was assessed using HeLa lysate or RRL and luciferase reporters, and the effect of the aptamer on factor association with eIF1A was analyzed by pull-downs and western blots.

## Anti-eIF1A aptamer transcription and eIF1A depletion

DNA encoding the $\alpha$-eIF1A aptamer sequence and the anti-sense control aptamer (AS) were generated using overlapping DNA oligonucleotides. RNA was generated by T7 transcription and folded in the presence of 2 mM Mg(OAc)$_2$. Aptamer at 3 µM final concentration was used in translation lysates and at 4 µM in PIC treatment. Samples were incubated for 15 min at either 30 or 37°C.

## Recombinant human eIF1A expression and purification

Human eIF1A was expressed from pET28-b in BL21 Star (DE3) *E. coli* (Invitrogen) and purified by Ni-NTA resin and gel filtration. Mutants of eIF1A contained: WT aa1-143, ΔNTT aa21-143, and ΔCTT aa1-117.

## Aptamer-depleted in vitro translation assays

RRL translation mixture contained 80% lysate (thawed on ice) supplemented with 5% 1 mM amino acid mixture, 10% 45 μM RNA aptamer (negative control or α-eIF1A aptamer) and 5% water. This mixture was incubated at 30°C for 15 min. The resulting aptamer-treated lysate mix was then added 2:1 to 18 μM eIF1A WT, ΔNTT, ΔCTT, or a buffer-only control. Depleted lysate with added-back eIF1A was then used to translate bicistronic reporter mRNA for 15 min at 30°C. Samples were analyzed for luciferase activity. Raw luciferase values were reported as ratios of α-eIF1A aptamer-treated translation to negative control aptamer-treated translation.

## Native PIC purification

293F cells were grown in suspension and left untreated or pretreated with 2 mM DTT before pelleting. Pellets were resuspended in 1X Ribo A (20 mM HEPES-KOH (pH 7.4), 100 mM KOAc (pH 7.6), 2.5 mM Mg(OAc)$_2$) and lysed by 3X freeze/thaw and passed through a 26G needle, then clarified by centrifugation. Lysate was left untreated on ice or incubated with 1.5 mM GMP-PNP. Ribosomes were pelleted by a 40,000 rpm spin in a Ti 50.2 rotor for 4 hr, resuspended in buffer A (20 mM Tris-HCl (pH 7.5), 50 mM KCl, 4 mM MgCl$_2$, 2 mM DTT), then ultracentrifuged through a 10–40% sucrose gradient in buffer B (20 mM Tris-HCl (pH 7.5), 25 mM KCl, 3 mM MgCl$_2$, 2 mM DTT) for 4 hr at 36,000 rpm in a SW41 rotor. 40S-containing fractions were collected, concentrated and stored at −80°C in buffer C (20 mM Tris-HCl (pH 7.5), 25 mM KCl, 3 mM MgCl$_2$, 2 mM DTT, 0.1 mM EDTA, 0.25 M sucrose).

## IRES:43S complex formation and sucrose density centrifugation

Purified native PICs were thawed on ice, incubated with WT or mutant IRES for 15 min at 37°C, then resolved by sucrose gradient ultracentrifugation. Gradients were fractionated, and then fractions containing 40S-bound or unbound peaks were collected separately, concentrated, and used in western or northern blotting. See *Figure 4—figure supplement 1* and *Figure 5—figure supplement 1*.

## Northern dot-blotting and analysis

Sucrose gradient analysis on remodeled 43S complexes was performed similarly as above except complexes were pretreated for 10 min at 37°C with aptamer. Ribosome-associated fractions were pooled and total RNA was fixed onto a nylon membrane. Crosslinked blots were first equilibrated, then hybridized in hyb buffer with 100,000–400,000 cpm 5' radiolabeled DNA oligos at 60°C overnight. Blots were washed with hyb buffer, exposed to a phosphorscreen, and analyzed using Image-Quant software.

## Statistics

Statistics for translation data and northern blotting were calculated in GraphPad Prism software. Data were represented as the average of n $\geq$ 3 $\pm$ SEM. Statistical validation was done using two-way analysis of variance (ANOVA) followed by a post-test using the Bonferroni method. Statistical significance shown by: *$p < 0.05$, **$p < 0.01$, ***$p < 0.001$.

## Acknowledgements

The authors thank Dr. Megan Filbin-Wong, Dr. Thomas Morrison and David Costantino for critical reading of the manuscript and insightful comments, and current and former Kieft Lab members for thoughtful discussions and technical assistance. We thank T. Pestova and C. Hellen for the eIF1A expression vector and vectors encoding various type 3 IRESs. The pDBS vector was generated from pRL, a gift of A. Willis. This work was supported by NIH grants GM081346 and GM118070 to JSK. JSK was previously an Early Career Scientist of the Howard Hughes Medical Institute.

## Additional information

### Funding

| Funder | Grant reference number | Author |
|---|---|---|
| National Institutes of Health | GM081346 | Jeffrey S Kieft |
| Howard Hughes Medical Institute | Early Career Scientist Award | Jeffrey S Kieft |
| National Institutes of Health | GM118070 | Jeffrey S Kieft |

The funders had no role in study design, data collection and interpretation, or the decision to submit the work for publication.

### Author contributions

ZAJ, Conceptualization, Formal analysis, Validation, Investigation, Methodology, Writing—original draft, Writing—review and editing; AO, YN, Resources, Formal analysis, Validation, Investigation, Methodology, Writing—review and editing, Developed and validated the aptamer technology that provided a critical method and resource for this study; JSK, Conceptualization, Resources, Formal analysis, Supervision, Funding acquisition, Project administration, Writing—review and editing

### Author ORCIDs

Jeffrey S Kieft, http://orcid.org/0000-0002-3718-1891

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
