## [Decision Letter]

Thank you for submitting your article "Bacterial-like translation initiation by the hepatitis C virus IRES requires eIF1A and ribosomal complex remodeling" for consideration by *eLife*. Your article has been reviewed by three peer reviewers, one of whom is a member of our Board of Reviewing Editors, and the evaluation has been overseen by James Manley as the Senior Editor. The reviewers have opted to remain anonymous.

The reviewers have discussed the reviews with one another and the Reviewing Editor has drafted this decision to help you prepare a revised submission.

This is an interesting manuscript by Jaafar and colleagues who re-examine the models of translation by the HCV IRES with a lysate-based system, finding that the factor eIF1A (homolog of bacterial IF1) is surprisingly required for efficient translation by type 3 IRESs. They additionally argue that other factors suggested to impact this IRES-mediated reaction, eIF2A and eIF2D, play no critical role in this process, and that tRNA can be loaded independently of eIF2A in a fashion that in some ways reflects a more "bacterial" event. They also argue that the IRES remodels PICs rather than binding the naked 40S subunit in the first step of initiation. Overall, the three reviewers were broadly enthusiastic, though as a group, it was felt that certain critical controls and quantification were lacking, and that conclusions in a number of places needed to be more cautious.

The main issues had first to do with the possibility that additional components (eIF2A, eIF3 or tRNA) were removed by the aptamer during the depletion experiments, and whether that explains the fact that activity was not fully restored when eIF1A was added back. Other concerns related to the absence of quantification of western blots throughout and to over-simplifications of more complex data (e.g. reviewer 2’s second comment concerning the impact of eIF1A on three different viruses HCV, CSFV and SPV-1). Finally, while several reviewers appreciated the comparison of the IRES system to that of bacteria (in its reliance on eIF1A and an eIF2A-independent tRNA loading step), reviewer 3 felt that this parallel was overstated given the lack of a role for eIF1 and the contributions of eIF3. As such, this story line should be eliminated or minimally more carefully states. Finally, reviewer 3 had substantial concerns about DTT vs. thapsigargin stress and the lack of a positive control for these stress conditions.

In closing, the manuscript is of sufficient interest for publication in *eLife* if the critical concerns of the reviewers (summarized above and detailed in the individual reviews) are adequately addressed.

Reviewer #1:

This manuscript describes an unanticipated role for eIF1A in IRES dependent translation of the hepatitis C virus, in the presence and absence of stress-induced phosphorylation (and thus inactivation) of eIF2α. Using a biotinylated HCV IRES pull down assay (in extracts), the authors show that eIF1A is bound to the IRES along with eIF2α. To study the function of bound eIF1A, they developed an aptamer that binds eIF1A and thus can deplete this factor from extracts or from the IRES. With an IRES-driven luciferase reporter, the authors were then able to detect a significant decrease in the efficiency of translation upon aptamer treatment. When purified eIF1A was then added back to lysates previously treated with the aptamer, translation was largely (though not completely) restored. Furthermore, under stress conditions in which eIF2α was phosphorylated and thus not functional in initiation, IRES-mediated translation was increased. The authors argue that this reflected an increase in overall free initiation factors in the cell and the ability of the HCV IRES to bypass eIF2α function via increased reliance on eIF1A. These data together are reminiscent of a bacterial model for translation initiation that lacks an eIF2α homolog, and is thought instead to directly load initiator tRNA into the P site with the guidance of IF1. While the proposed mechanism of bacteria-like eIF1A-driven initiation gives potential insight into a novel role for eIF1A for certain viral IRES's as well as helping to reconcile conflicting models about how the IRES and the ribosome engage one another in the cell (factors first or IRES first), more evidence is needed to fully support the authors' claims. In particular, controls should be done to ensure off target effects are not responsible for certain results and the western blot data should be looked at or explained more carefully when analyzing results.

Major points:

1) eIF2α appears to be pulled down with eIF1A-aptamer complex, in addition to eIF1A (Figure 2—figure supplement 1). This prompts the question of whether the results shown with the aptamer are the result of the depletion of eIF1A or eIF2α from IRES or various extracts, or both. Shown in Figure 2 is a depletion of eIF1A from IRESs through an IRES pull-down experiment, mediated by the aptamer. A similar experiment probing for eIF2α would be sufficient to prove the aptamer results reflected the consequence of eIF1A depletion but not that of eIF2α depletion.

2) The authors state in paragraph two “HCV IRES RNA manipulates preassembled PICs and stabilizes eIF1A binding” that the addition of WT IRES caused a displacement of eIF2α in stressed and unstressed cells, while eIF3 and eIF1A remained bound. However, 1A seems to be displaced to the same extent that eIF2α was displaced in both conditions. Thus, when eIF2α was not present, eIF1A was also not present (Figure 4 lanes 1, 2, 7, 8); as such, these data are not supportive of an eIF1A-driven, eIF2α-independent IRES initiation model, as proposed.

Reviewer #2:

The manuscript entitled "Bacterial-like translation initiation by the hepatitis C virus IRES requires eIF1A and ribosomal complex remodeling" by Jaafar et al., (2016) reports on the investigation of the molecular mechanism of HCV IRES-driven translation initiation. In this work, the authors used a variety of in vitro approaches to identify the eukaryotic initiation factor (eIF) composition of pre-initiation complexes (PICs) assembled on the HCV IRES and to determine the functional significance of a subset of these factors using mRNA reporter-based activity assays. The authors demonstrate that eIF1A is a component of the HCV IRES-based PIC and, importantly, that this factor is required for efficient HCV IRES-driven translation initiation. In addition, the authors demonstrate that eIF1A plays a role in stabilizing the binding of the initiator tRNA (Met-tRNA_i_) at a canonical AUG start codon. Finally, the authors demonstrate that the knockdown of eIF2D and eIF2A, two factors previously implicated in Met-tRNA_i_ delivery under stress conditions, has no effect on the translation of a luciferase reporter driven by the HCV IRES. Based on their findings, the authors propose a model for HCV-IRES mediated translation initiation under normal and stress conditions. In this model, the HCV IRES binds and remodels native, pre-assembled PICs. The design of the experiments in this study is appropriate and with a few exceptions described below, the data, the data analysis, and the conclusions that are drawn from this data are acceptable. Overall, the results are novel and expand our understanding of how viral IRES elements co-opt the translational machinery to drive translation of viral RNAs. Assuming that the authors can appropriately address the major comments below and revise the manuscript accordingly, I highly recommend this manuscript for publication in *eLife*.

Major Comments

1) Throughout the manuscript, the analysis of the western blot data are described and presented qualitatively. The authors should provide a more quantitative analysis by measuring band intensities, normalizing to the loading control (rpS6) and reporting fold changes.

2) The authors write: "Examination of the effect of eIF1A depletion on HCV, CSFV, and SPV-1 IRESs revealed a functional requirement for eIF1A in all three." However, Figure 2 clearly shows that there is no effect of depleting eIF1A on SPV-1 driven luciferase translation in HeLa cells. The authors should clarify their sentence and comment on this discrepancy.

3) In several instances in the manuscript the authors state that deletion of the N-terminal tail of eIF1A lowers stringency followed by a reference for Fekete et al. (2007). However, in Fekete et al.,the authors demonstrate that point mutations in the CTT of eIF1A reduce the stringency of start codon selection (Sui- phenotype) and thus result in increased initiation at a UUG codon. In addition, they showed that mutations in the NTT of eIF1A suppressed the Sui- phenotype (Ssu- phenotype) suggesting that NTT mutants of eIF1A reduce initiation at UUG codons. Thus, rather than supporting the authors' results and conclusions, this previously published article seems to contradict them. The authors should revise the appropriate sections of the manuscript.

4) Regarding the data in Figure 3, the authors indicate that the activity of the IRESs seems to depend on the presence of the NTT to a greater extent than does cap-dependent translation. This statement does not appear to be justified by the data.

5) The authors propose a model in which the HCV IRES binds a pre-assembled PIC and alters its composition so as to favor 80S ribosome formation. However, it is also possible that the IRES simply binds to those PICs from which factors that are not present in the final HCV IRES-PIC have stochastically dissociated during incubation with the IRES RNA. The authors should acknowledge this possibility.

6) Regarding the data in Figure 4 with the (Δ)dII HCV IRES: In unstressed 293F cells PIC assembly with the (Δ)dII HCV IRES results in less eIF1A retained. However, this is not true of eIF1A in unstressed HeLa cells. Could the authors comment on this discrepancy?

7) eIF5 is an important component of the 43S IC that functions as a GTPase activating protein for eIF2, and also interacts with eIF3 and eIF1A during canonical translation initiation. Is there a reason why the authors did not probe for eIF5 in their PICs?

Reviewer #3:

The HCV IRES is a model type 3 IRES that uses a subset of canonical eukaryotic translation initiation factors (eIFs) and can operate under normal conditions and under stress, when eIF2 has been inactivated by phosphorylation of the α subunit. Several previous studies have defined HCV IRES eIF requirements to include eIF2, eIF3, eIF5, and eIF5B under normal condition, while there are conflicting reports about factor requirement in the absence of eIF2. Jaafar and colleagues re-examine the models of translation by the HCV IRES with a lysate-based system, finding that the factor eIF1A (homolog of bacterial IF1) is surprisingly also required for efficient translation by type 3 IRESs. Using RNAi-mediated knockdowns they further show that factors previously suggested to substitute for eIF2 under stress don't contribute to efficient translation, and argue that tRNA recruitment occurs without a specific factor, as is the case in bacteria. They conclude that HCV IRES translation is thus bacterial-like and requires remodeling of a pre-assembled canonical pre-initiation complex (PIC).

This article’s work around eIF1A is quite convincing in showing the unexpected role of the factor in translation by the HCV IRES. eIF1A is not just a component of the HCV IRES PIC, but it removal reduces translation activity in lysate, and restoring the factor with purified protein appears to restore this loss. However, it is worth noting that in many cases the factor add-back is incomplete (Figure 2) suggesting that the anti-eIF1A aptamer might remove additional factors and/or tRNA that are important for IRES function. It would be important to clarify this, by blotting against the aptamer what the aptamer depleted, possibly with a pull down experiment. Prime candidates would be eIF3 (which binds eIF1A and could be pulled down alongside) and tRNA_i_ which might interact with the aptamer (which has an AUG in a stem loop). In the former case, eIF3 itself may bind the aptamer, due to its intrinsic RNA binding activity, and be depleted. In the latter case, it'd be important to control for tRNA loss given the result presented in Figure 5—figure supplement 1, where the aptamer appears to reduce the ratio of tRNA:18S rRNA. Moreover, several blots (Figure 4 and Figure 5) appear to be underexposed and lack important positive controls that would ensure that the experiment would be detecting protein if present. In their assessment of the role of eIF2A and eIF2D, I am concerned about the use of thapsigargin as the stressor and the lack of a positive control for inactivating the IRES under stress (Figure 5—figure supplement 2). The authors also use DTT as a stressor in separate experiments, and don't explain why thapsigargin was used in these experiments. To make these experiments conclusive, it would be important to demonstrate that eIF2α is phosphorylated under these conditions and the inclusion of a knockdown that actually reduces luciferase activity in Figure 5—figure supplement 2. Given that thapsigargin raises intracellular Ca^2+^ levels, this chemical might allow for a completely factor-independent mode of IRES translation, as seen with high Ca^2+^ or Mg^2+^ (Lancaster et al. 2006). The authors make the case that the IRES remodels PICs rather than binding the naked 40S subunit in the first step of initiation. While this is certainly plausible, and it is possible that eIF1A stays bound following IRES binding, their data does not show that the IRES actively remove eIF3 to position it at a new location (eIF3 could simply disassociate and rebind, since their experiments are not done under dilute conditions). Given that this conclusion is featured prominently in their title, Abstract, Discussion, and Figure 6, it is important that the authors either provide data in support of this conclusion or clarify that it is only speculation. Contrary to what the authors have stated (Discussion paragraph seven) the IRES does not have much similarity in contacts with eIF3 and does eIF4G, as the former interacts with the IRESs predominantly through the subunit a and c, while the latter interacts with subunits c, d, and e (per their references). This comparison has been made previously by other (Siridechadilok et al. 2005) and is clearly incompatible based on current structural models. Finally, while aspects of the IRES translation might be considered "bacterial-like" this comparison is generally out of place, especially given the lack of a SD sequence, the absence of eIF1 function (IF3 analog), and the utilization of eIF3, which is nowhere in bacteria.

Although the eIF1A data is fairly convincing, this article needs serious revisions to guarantee that the conclusions are not beyond the scope of their experiments. Provided that the reviewers can agree on a short list of experiments, this article could eventually be published in *eLife* provided the necessary changes are made.

---

## [Author Response]

*This is an interesting manuscript by Jaafar and colleagues who re-examine the models of translation by the HCV IRES with a lysate-based system, finding that the factor eIF1A (homolog of bacterial IF1) is surprisingly required for efficient translation by type 3 IRESs. They additionally argue that other factors suggested to impact this IRES-mediated reaction, eIF2A and eIF2D, play no critical role in this process, and that tRNA can be loaded independently of eIF2A in a fashion that in some ways reflects a more "bacterial" event. They also argue that the IRES remodels PICs rather than binding the naked 40S subunit in the first step of initiation. Overall, the three reviewers were broadly enthusiastic, though as a group, it was felt that certain critical controls and quantification were lacking, and that conclusions in a number of places needed to be more cautious.*

*The main issues had to do first with the possibility that additional components (eIF2A, eIF3 or tRNA) were removed by the aptamer during the depletion experiments, and whether that explains the fact that activity was not fully restored when eIF1A was added back.*

We agree that this is an important control for eIF2 and eIF3. However, checking for direct aptamer interaction with tRNA_i_ is confounded by the fact that we expect tRNA_i_ to no longer bind to the PIC if eIF1A is depleted. This is the conclusion we make from the data presented in Figure 5 and Figure 5—figure supplement 1. In other words, tRNA_i_ loss is not a control, it is an experimental result that supports our model.

We have added this control to Figure 2. The results show that the occupancy of other needed factors (eIF2, eIF3) is not affected by the aptamer.

*Other concerns related to the absence of quantification of western blots throughout and to over-simplifications of more complex data (e.g. reviewer 2’s second comment concerning the impact of eIF1A on three different viruses HCV, CSFV and SPV-1).*

We were hesitant to include quantitative data, as these experiments were designed to be qualitative in nature. Quantitation between experiments is tricky, even when the trends are clear. However, we agree that this is helpful in supporting our data in key places, and have added quantitation where appropriate.

We include quantitated and normalized data from multiple blots in a graph form in Figure 1. To avoid complicating Figure 4, we have included quantitated data in tabular form in Figure 4—figure supplement 2.

As described in more detail in the responses to the individual reviewers, we have added discussion in several places in the manuscript to clarify and avoid oversimplification.

*Finally, while several reviewers appreciated the comparison of the IRES system to that of bacteria (in its reliance on eIF1A and an eIF2A-independent tRNA loading step), reviewer 3 felt that this parallel was overstated given the lack of a role for eIF1 and the contributions of eIF3. As such, this story line should be eliminated or minimally more carefully states.*

We point out that the absence of a role for eIF1 is, in fact, fully consistent with bacterial initiation in that eIF1’s homolog, IF3, is not universally present in all bacterial species. Also, the role of eIF3 is still poorly defined; it is possible that eIF3 is needed only to prepare the PIC during recycling and not for subsequent HCV IRES-specific steps. Nonetheless, we agree that this conclusion could be tempered.

We have adjusted the manuscript to not overstate this point, to include altering the title.

*Finally, reviewer 3 had substantial concerns about DTT vs. thapsigargin stress and the lack of a positive control for these stress conditions.*

We agree, there were a few important controls that were conducted but not included in the original manuscript. We have added these.

We have added additional experiments with DTT to parallel the thapsigargin treatment; these are included in Figure 5—figure supplement 2. Controls in the form of detection of eIF2-α phosphorylation and translation of eIF2-dependent or eIF2-independent messages are in Figure 4—figure supplement 1.

*Reviewer #1:*

*[…]Major points:*

*1) eIF2α appears to be pulled down with eIF1A-aptamer complex, in addition to eIF1A (Figure 2—figure supplement 1). This prompts the question of whether the results shown with the aptamer are the result of the depletion of eIF1A or eIF2α from IRES or various extracts, or both. Shown in Figure 2 is a depletion of eIF1A from IRESs through an IRES pull-down experiment, mediated by the aptamer. A similar experiment probing for eIF2α would be sufficient to prove the aptamer results reflected the consequence of eIF1A depletion but not that of eIF2α depletion.*

We agree that this is an important control. We have added this control to Figure 2. The results show that the occupancy of other needed factors (eIF2, eIF3) is not affected by the aptamer at the stated concentration.

*2) The authors state in paragraph two “HCV IRES RNA manipulates preassembled PICs and stabilizes eIF1A binding”, that the addition of WT IRES caused a displacement of eIF2α in stressed and unstressed cells, while eIF3 and eIF1A remained bound. However, 1A seems to be displaced to the same extent that eIF2α was displaced in both conditions. Thus, when eIF2α was not present, eIF1A was also not present (Figure 4 lanes 1, 2, 7, 8); as such, these data are not supportive of an eIF1A-driven, eIF2α-independent IRES initiation model, as proposed.*

We disagree with the reviewer on this point. In the stressed cells (Figure 4), eIF2α is undetected in all lanes, but eIF1A is clearly still bound to similar levels both in the presence and absence of HCV IRES. In the absence of stress, eIF2α is present but its occupancy is decreased when the IRES binds, but eIF1A remains bound. Hence, the blots show independence of these two factors.

We have not made specific changes, as we feel the observations are correct as reported.

*Reviewer #2:*

*[…] Major Comments*

*1) Throughout the manuscript, the analysis of the western blot data are described and presented qualitatively. The authors should provide a more quantitative analysis by measuring band intensities, normalizing to the loading control (rpS6) and reporting fold changes.*

We were hesitant to include quantitative data; these experiments were designed to be qualitative in nature. Quantitation between experiments is tricky, even when the trends are clear. However, we agree that this is helpful in supporting our data in key places, and have added quantitation where appropriate.

We include quantitated and normalized data from multiple blots in a graph form in Figure 1. To avoid complicating Figure 4, we have included quantitated data in tabular form in Figure 4—figure supplement 2.

*2) The authors write: "Examination of the effect of eIF1A depletion on HCV, CSFV, and SPV-1 IRESs revealed a functional requirement for eIF1A in all three." However, Figure 2 clearly shows that there is no effect of depleting eIF1A on SPV-1 driven luciferase translation in HeLa cells. The authors should clarify their sentence and comment on this discrepancy.*

We agree this this sentence oversimplified the result. The SPV-1 IRES shows some decrease in activity in RRL, but not to the degree that HCV and CSFV do. It shows little or no decrease in HeLa lysate. Interestingly, the SPV-1 IRES pulls down less eIF1A (Figure 1). While we lack the data to present a detailed mechanistic explanation for these results, the data we do have speak to the idea that different type 3 IRESs have mechanistic variation. We have added a clearer description and explanation.

*3) In several instances in the manuscript the authors state that deletion of the N-terminal tail of eIF1A lowers stringency followed by a reference for Fekete et al. (2007). However, in Fekete et al., the authors demonstrate that point mutations in the CTT of eIF1A reduce the stringency of start codon selection (Sui- phenotype) and thus result in increased initiation at a UUG codon. In addition, they showed that mutations in the NTT of eIF1A suppressed the Sui- phenotype (Ssu- phenotype) suggesting that NTT mutants of eIF1A reduce initiation at UUG codons. Thus, rather than supporting the authors' results and conclusions, this previously published article seems to contradict them. The authors should revise the appropriate sections of the manuscript.*

The roles for the N and C terminal tails of eIF1A are many and complex, and we agree that we did a poor job of explaining this and how it supports our data. We point out that while Fekete et al. (2007) do report a “hyperaccurate” phenotype when they mutate parts of the NTT, it is not clear that complete deletion of the tail would have the same effect, especially without other mutations also present.

We have altered the language to be clearer about the possible interpretations of our results in light of the existing data on the tails of eIF1A

*4) Regarding the data in Figure 3, the authors indicate that the activity of the IRESs seems to depend on the presence of the NTT to a greater extent than does cap-dependent translation. This statement does not appear to be justified by the data.*

This statement was based on the observation that while addition of △NNT-eIF1A to lysate depleted of eIF1A led to a full restoration of cap-dependent translation, it did not fully restore IRES-driven translation. However, we agree that there is some variability in the effect, and in some cases the add-back of △NNT-eIF1A and WT-eIF1A give similar effects. Overall, we agree that our statement may lead to some confusion; certainly it is not essential for the conclusions. We have removed this statement.

*5) The authors propose a model in which the HCV IRES binds a pre-assembled PIC and alters its composition so as to favor 80S ribosome formation. However, it is also possible that the IRES simply binds to those PICs from which factors that are not present in the final HCV IRES-PIC have stochastically dissociated during incubation with the IRES RNA. The authors should acknowledge this possibility.*

We agree that we cannot completely exclude this possibility. However, the inclusion of a no-RNA control helps to address this. If factors tend to stochastically dissociate, they likely would not be present on the no-RNA control PICs, which were subjected to the same incubation and centrifugation as those with IRES. Any effects we see are due to the IRES, so even if factors do stochastically dissociate during incubation, the IRES would have to act to block their re-binding; essentially this is a form of remodeling to alter the composition of the PIC, albeit perhaps by a subtly different pathway. We have added a statement to address this possibility.

*6) Regarding the data in Figure 4 with the (Δ)dII HCV IRES: In unstressed 293F cells PIC assembly with the (Δ)dII HCV IRES results in less eIF1A retained. However, this is not true of eIF1A in unstressed HeLa cells. Could the authors comment on this discrepancy?*

We agree that the data indicate that △dII, with PICs from HeLa cells, does not show the same effect as with unstressed 293 cells. The reason is unclear, but could be due to subtle differences in the makeup or configuration of the PIC from these cells. Without deeper analysis, we are hesitant to ascribe a reason or significance to this, but we agree that it should be noted. We have added text noting and commenting on this observation.

*7) eIF5 is an important component of the 43S IC that functions as a GTPase activating protein for eIF2, and also interacts with eIF3 and eIF1A during canonical translation initiation. Is there a reason why the authors did not probe for eIF5 in their PICs?*

The referee’s point is well taken as eIF5 is indeed a component of the 43S PIC. Though eIF5 is known to interact with multiple components of the PIC and the multifactor complex, its only known functions are to direct recruitment and GTPase activation of eIF2. We have examined the occupancy of eIF2 in our data; we feel the presence of eIF2 is sufficient to determine whether eIF5 could be functionally relevant. Further, Western blotting we have performed regarding the presence of eIF5 on IRES pull-down experiments do not detect eIF5 on the complexes pulled out of lysate. We have chosen not to include that information in order to avoid complicating Figure 1, Figure 4, and 4—figure supplement 1 with negative data.

*Reviewer #3:*

*[…] This article’s work around eIF1A is quite convincing in showing the unexpected role of the factor in translation by the HCV IRES. eIF1A is not just a component of the HCV IRES PIC, but it removal reduces translation activity in lysate, and restoring the factor with purified protein appears to restore this loss. However, it is worth noting that in many cases the factor add-back is incomplete (Figure 2) suggesting that the anti-eIF1A aptamer might remove additional factors and/or tRNA that are important for IRES function. It would be important to clarify this, by blotting against the aptamer what the aptamer depleted, possibly with a pull down experiment. Prime candidates would be eIF3 (which binds eIF1A and could be pulled down alongside) and tRNA_i_ which might interact with the aptamer (which has an AUG in a stem loop). In the former case, eIF3 itself may bind the aptamer, due to its intrinsic RNA binding activity, and be depleted. In the latter case, it'd be important to control for tRNA loss given the result presented in Figure 5—figure supplement 1, where the aptamer appears to reduce the ratio of tRNA:18S rRNA.*

We agree that checking for an aptamer-induced loss of eIFs 2 and 3 is an important control. However, checking for direct aptamer interaction with tRNA_i_ is confounded by the fact that we expect tRNA_i_ to no longer bind to the PIC if eIF1A is depleted. This is the conclusion we make from the data presented in Figure 5 and Figure 5—figure supplement 1. In other words, tRNA_i_ loss is not a control, it is an experimental result that supports our model.

We have added a control to Figure 2. The results show that the occupancy of other needed factors (eIF2, eIF3) is not affected by the aptamer.

*Moreover, several blots (Figure 4 and Figure 5) appear to be underexposed and lack important positive controls that would ensure that the experiment would be detecting protein if present.*

In fact, these blots were exposed to a degree that even residual protein would be seen. Every protein that we blot for is detected on one of the two panels, which represent a single experiment.

*In their assessment of the role of eIF2A and eIF2D, I am concerned about the use of thapsigargin as the stressor and the lack of a positive control for inactivating the IRES under stress (Figure 5—figure supplement 2). The authors also use DTT as a stressor in separate experiments, and don't explain why thapsigargin was used in these experiments.*

Our choice of thapsigargin was made to exactly replicate previously published results exploring eIF2A and 2D (Kim et al., 2011). However, to better explore this, we repeated the experiment with DTT. With these experiments, we also included eIF1A knockdown as a positive control for HCV IRES inhibition under stress.

We have added the results of these experiments to Figure 5—figure supplement 2

*To make these experiments conclusive, it would be important to demonstrate that eIF2α is phosphorylated under these conditions and the inclusion of a knockdown that actually reduces luciferase activity in Figure 5—figure supplement 2. Given that thapsigargin raises intracellular Ca^2+^ levels, this chemical might allow for a completely factor-independent mode of IRES translation, as seen with high Ca^2+^ or Mg^2+^ (Lancaster et al. 2006).*

These are good controls, and we agree that we should have included them. The treatment with thapsigargin is unlikely to switch the IRES into this mode, given the very special conditions needed to achieve 80S formation on the IRES with no factors in vitro. However, the eIF1A control now accounts for this, showing that under stress, the IRES still requires this factor.

We have included a blot showing induced eIF2α phosphorylation induced by different stressors in Figure 4—figure supplement 1, as well as controls for the effect on cap-dependent translation and two other IRESs known to be eiF2-dependent or eIF2-independent in Figure 4—figure supplement 1. Also, as mentioned above, Figure 5—figure supplement 2 contains the results of an eIF1A knockdown under stress.

*The authors make the case that the IRES remodels PICs rather than binding the naked 40S subunit in the first step of initiation. While this is certainly plausible, and it is possible that eIF1A stays bound following IRES binding, their data does not show that the IRES actively remove eIF3 to position it at a new location (eIF3 could simply disassociate and rebind, since their experiments are not done under dilute conditions). Given that this conclusion is featured prominently in their title, Abstract, Discussion, and Figure 6, it is important that the authors either provide data in support of this conclusion or clarify that it is only speculation.*

This matches the point of Reviewer 2, point 5, above. As stated above, we agree that we cannot completely exclude this possibility. However, by comparing the No RNA and WT IRES lanes, it is clear that the effects we see are due to the IRES. If the IRES actively dissociates eIF3 during incubation then the factor rebinds in a new way; we would say that this is still a form of remodeling, albeit perhaps by a subtly different pathway. We have added a statement to address this possibility.

*Contrary to what the authors have stated (Discussion paragraph seven) the IRES does not have much similarity in contacts with eIF3 and does eIF4G, as the former interacts with the IRESs predominantly through the subunit a and c, while the latter interacts with subunits c, d, and e (per their references). This comparison has been made previously by other (Siridechadilok et al. 2005) and is clearly incompatible based on current structural models.*

We agree that this may be over-reaching a bit. We have removed this statement.

*Finally, while aspects of the IRES translation might be considered "bacterial-like" this comparison is generally out of place, especially given the lack of a SD sequence, the absence of eIF1 function (IF3 analog), and the utilization of eIF3, which is nowhere in bacteria.*

We point out that the absence of a role for eIF1 is, in fact, fully consistent with bacterial initiation in that eIF1’s homolog, IF3, is not universally present in all bacterial species. Also, the role of eIF3 is still poorly defined; it is possible that eIF3 is needed only to prepare the PIC during recycling and not for subsequent HCV IRES-specific steps. We do not claim that IRES recruitment of the small subunit is bacterial-like (although this analogy has been made by others), but that tRNA binding has aspects similar to that in bacteria. Nonetheless, we agree that this conclusion could be tempered.

We have adjusted the manuscript to not overstate this point, to include altering the title.